# A distinct isoform of ZNF207 controls self-renewal and pluripotency of human embryonic stem cells

Fang Fang [1,2], Ninuo Xia[1,2], Benjamin Angulo[1,2], Joseph Carey[1,2], Zackery Cady[1,2], Jens Durruthy-Durruthy[3], Theo Bennett[1,2], Vittorio Sebastiano[3] & Renee A. Reijo Pera [1,2]

Self-renewal and pluripotency in human embryonic stem cells (hESCs) depends upon the function of a remarkably small number of master transcription factors (TFs) that include OCT4, SOX2, and NANOG. Endogenous factors that regulate and maintain the expression of master TFs in hESCs remain largely unknown and/or uncharacterized. Here, we use a genome-wide, proteomics approach to identify proteins associated with the *OCT4* enhancer. We identify known OCT4 regulators, plus a subset of potential regulators including a zinc finger protein, ZNF207, that plays diverse roles during development. In hESCs, ZNF207 partners with master pluripotency TFs to govern self-renewal and pluripotency while simultaneously controlling commitment of cells towards ectoderm through direct regulation of neuronal TFs, including OTX2. The distinct roles of ZNF207 during differentiation occur via isoform switching. Thus, a distinct isoform of ZNF207 functions in hESCs at the nexus that balances pluripotency and differentiation to ectoderm.

[1] Department of Cell Biology and Neurosciences, Montana State University, Bozeman, MT 59717, USA. [2] Department of Chemistry and Biochemistry, Montana State University, Bozeman, MT 59717, USA. [3] Department of Genetics, Institute for Stem Cell Biology and Regenerative Medicine, Stanford University School of Medicine, Stanford, CA 94305, USA. These authors contributed equally: Fang Fang, Ninuo Xia. Correspondence and requests for materials should be addressed to F.F. (email: fangfang0724@gmail.com)

Human embryonic stem cells (hESCs) possess the capacity to renew indefinitely (self-renewal) while maintaining the potential to differentiate into any somatic cell types (pluripotency). Self-renewal and pluripotency are regulated by a unique transcriptional network controlled by a small number of endogenous master transcription factors (TFs) that include OCT4, SOX2, and NANOG[1–5]. Disruption in the expression of critical TFs leads to loss of pluripotency and commitment of cells to differentiate into diverse cell lineages. For example, in mouse embryonic stem cells (mESCs), a 50% reduction in expression of OCT4 (also known as POU5F1) triggers mESC differentiation to trophoblasts; conversely, a twofold increase in expression promotes differentiation to primitive endoderm and mesoderm[6]. While multiple reports document the function of TFs via studies that modulate exogenous expression in pluripotent stem cells, much less is known about how TFs are endogenously regulated.

Recently, locus-specific proteomics makes use of the genomic targeting ability of the TALEN (Transcription Activator-Like Effector Nuclease) or CRISPR (Clustered Regularly Interspaced Palindromic Repeats) systems to achieve high resolution and specific isolation of factors that bind native genomic loci. It has the potential to provide insight into endogenous protein and epigenetic mechanisms that regulate gene expression[7–14]. However, these studies have never been done in hESCs.

Our study reports the adaptation and optimization of locus-specific proteomics to target the regulatory element of the OCT4 gene in hESCs. It is important to note that OCT4 gene expression is regulated by three regulatory elements: a distal enhancer (DE), a proximal enhancer (PE), and a proximal promoter (PP)[15–17]. The PE element is used in hESCs to maintain OCT4 expression[18]. The goal of this study was to identify nuclear proteins bound at the PE of OCT4, including most known OCT4 regulators as well as TFs that have not previously been shown to bind at this locus. Among the list of TFs, we identified a zinc finger protein, ZNF207. Zinc finger-containing proteins (ZNFs) are the largest transcription factor family in the human genome, where, in combination with other TFs or co-factors, ZNFs play diverse roles in development, differentiation, and metabolism. This study reports identification and characterization of the essential roles of ZNF207 in self-renewal, pluripotency, and differentiation.

## Results

**Locus-specific proteomics identified OCT4 regulators**. We set out to develop an unbiased, genome-wide method to screen for proteins that locate to the OCT4 PE in hESCs; for this purpose, we developed an optimized locus-specific proteomics approach in hESCs (Fig. 1a). First, we designed TALEN constructs to target the sequences that are near the OCT4 PE, located in a region of DNaseI hypersensitivity (Supplementary Fig. 1a). TALEN constructs with the highest cutting efficiency were chosen for locus-specific proteomics (Supplementary Table 1). We then made modifications to the original TALEN protein to transform it into a catalytically-dead TALE (dTALE) protein that is optimized for locus-specific proteomics in hESCs (Supplementary Fig. 1b) via three steps: (1) The nuclease-domain FokI at the C-terminus was replaced by a GFP (green fluorescence protein); (2) a 3X FLAG tag at the N-terminus was included for following pull-down analysis; (3) the existing CMV promoter was replaced with an EF1α promoter that has robust expression in hESCs (Supplementary Fig. 1c). This dTALE protein could then be chemically crosslinked to the OCT4 locus together with all the other proteins that bind to the locus. We verified that dTALE protein binds to the targeted locus by chromatin immunoprecipitation (ChIP)-qPCR (Fig. 1b). Following crosslinking, chromatin was sheared, and all the associated proteins were immunoprecipitated using an anti-FLAG antibody. Immunoprecipitation pulled down the dTALE protein (Supplementary Fig. 1d) as well as other proteins and complexes that are also attached to that region (Supplementary Fig.1e). Crosslinking was then reversed, and the samples were subjected to mass spectrometry to enable generation of a list of proteins that potentially bind to OCT4 PE locus.

Mass spectrometry resulted in the identification of 150 nucleus proteins co-immunoprecipitated with dTALE protein (Supplementary Data file). The list of proteins includes proteins that are known to bind to and regulate OCT4, including Sex Determining Region Y-Box 2 (SOX2), Spalt-Like Transcriptional Factor 4 (SALL4), Histone Deacetylase 2 (HDAC2), and OCT4 itself (Supplementary Table 2)[1,19–21]. The identification of these known regulators of OCT4 provides support for the reliability of our approach. In addition to these known OCT4 locus-bound proteins, mass spectrometry also generated a list of nuclear proteins, including a few TFs (Supplementary Table 3). Enriched Gene Ontology (GO) categories for these proteins were identified using DAVID (the Database for Annotation, Visualization and Integrated Discovery) (Fig. 1c). These proteins include those that are involved in transcriptional regulation, translational regulation, nuclear structure, as well as, nuclear transport machinery. This is consistent with the previous findings that the transcriptional machinery is coupled with the translational machinery and nuclear transport machinery at transcriptionally active sequences[22].

**ZNF207 is required for self-renewal and pluripotency**. Among the 150 nuclear proteins identified, we focused on TFs and identified a subset for further verification (Supplementary Table 3); all of these demonstrated binding at the PE of OCT4 by ChIP-qPCR (Supplementary Fig. 2a). OCT4 protein was used as the positive control. To test whether these TFs play regulatory roles at the OCT4 genetic locus, we performed siRNA-mediated knockdown (KD) analysis. Q-PCR results indicated that KD of several of these TFs led to a significant change in OCT4 mRNA expression. KD of MBD3, CBX3, IGF2BP1, and OCT4 caused reduction of OCT4 expression, while KD of HDAC2 and SALL2 resulted in upregulation of OCT4 expression (Supplementary Fig. 2b). Both downregulation and upregulation of OCT4 in hESCs results in differentiation. These results, consistent with the published data, suggest that OCT4 level must be tightly controlled in order to maintain self-renewal and pluripotency[6,23]. These results also demonstrated the ability of the optimized locus-specific proteomics method to identify putative regulators of OCT4 that may function in maintaining self-renewal and pluripotency of hESCs.

The KD of ZNF207 was notable as it triggered a dramatic morphological change in hESC colonies, even relative to the KD phenotype of OCT4 and SOX2 (Fig. 2a). The mRNA and protein levels of ZNF207 were reduced by siRNAs to about 5–10% (Supplementary Fig. 2c). The morphological changes were correlated with loss of alkaline phosphatase (AP) staining (Fig. 2b). Utilizing ChIP, we also verified that ZNF207 binds to the PE of OCT4 (Fig. 2c). As ZNF207 expression decreased, protein expression of OCT4 and NANOG was drastically reduced; moreover, the hESC surface marker TRA-1-60, changed its localization from even distribution on the cell membrane to a focal point inside the cell (Fig. 2d). These results indicated that hESCs have undergone dramatic changes, commonly associated with loss of pluripotency and self-renewal, in response to reduced expression of ZNF207. To further examine the role of ZNF207 in induction of the endogenous OCT4 gene, we compared the colony formation efficiency during the reprogramming process from human neonatal fibroblast to induced pluripotent stem cells

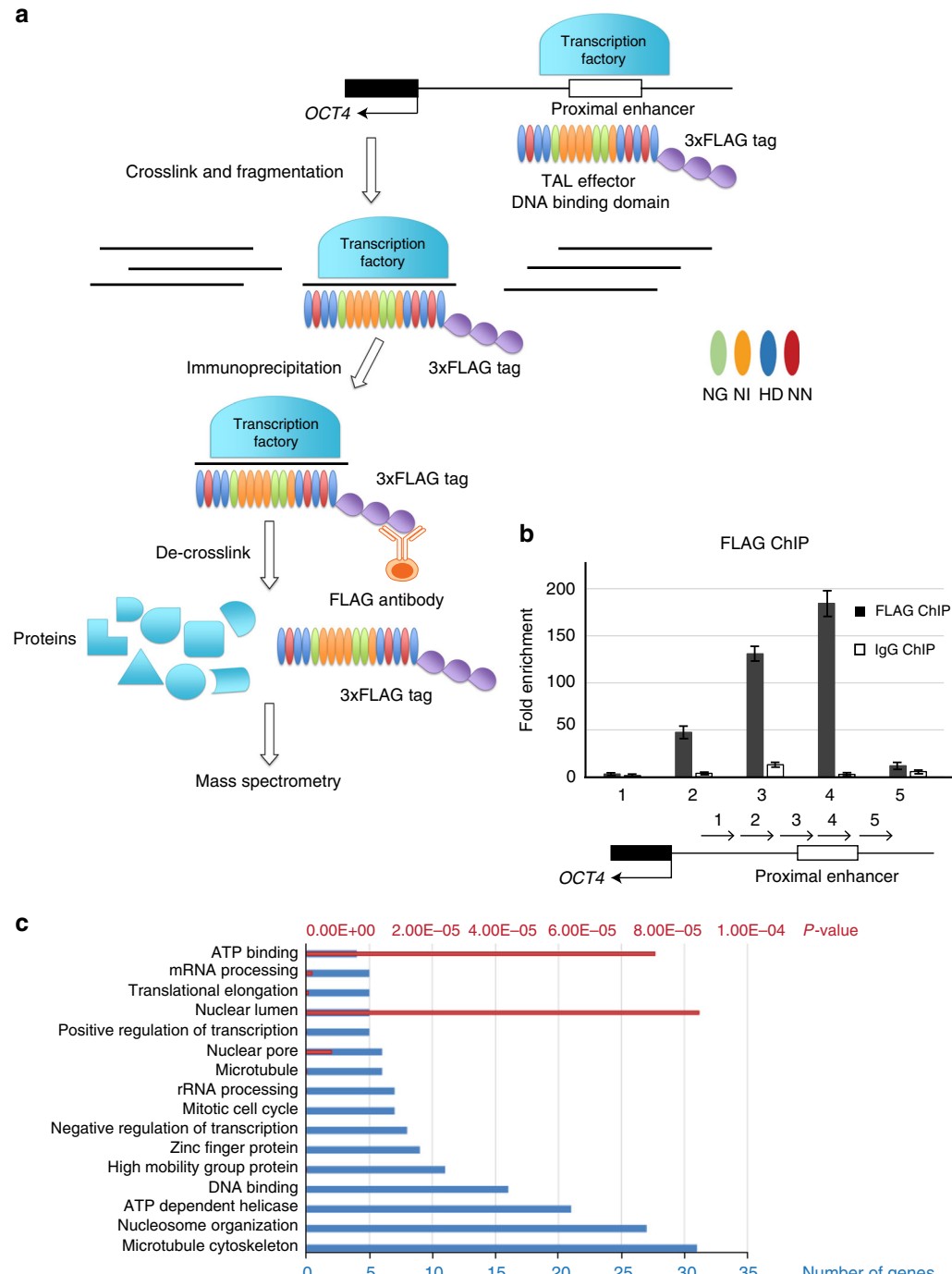

**Fig. 1** Locus-specific proteomics identified proteins located at the proximal enhancer of *OCT4* gene in hESCs. **a** Schematic overview of locus-specific proteomics in hESCs. A representation of *OCT4* locus is shown on the top. Dark boxes represent exons, and the white box represents the proximal enhancer that is bound by the transcription factory. TALEN protein with a 3xFLAG tag was designed to bind to the proximal enhancer. The colored ovals represent the "repeat-variable di-residues" (RVD) of TALEN protein that determines binding specificity to DNA bases. They follow the code that NG, NI, HD, and NN respectively recognizes thymine, adenine, cytosine, and guanine. Chromatin is crosslinked by formaldehyde and fragmented by sonication. Then, FLAG antibody was used for immunoprecipitation of proteins bound to the proximal enhancer. The pull-down complex was de-crosslinked of the identity of isolated proteins was discovered by mass spectrometry. **b** Validation of the binding of dTALE protein by ChIP. A ChIP assay was performed using anti-flag antibody to detect enriched fragments in hESCs. Fold enrichment is the relative abundance of DNA fragments at the amplified region over a control amplified region. IgG ChIP is served as control. The locations of the amplified products are indicated by arrows along the proximal promoter of *OCT4*. Data are presented as the mean ± SEM and are derived from three independent experiments. **c** Gene ontology analysis of the proteins identified by locus-specific proteomics. Blue bars represent the number of genes and red bars represent *p*-value. *p*-value is calculated using hypergeometric distribution

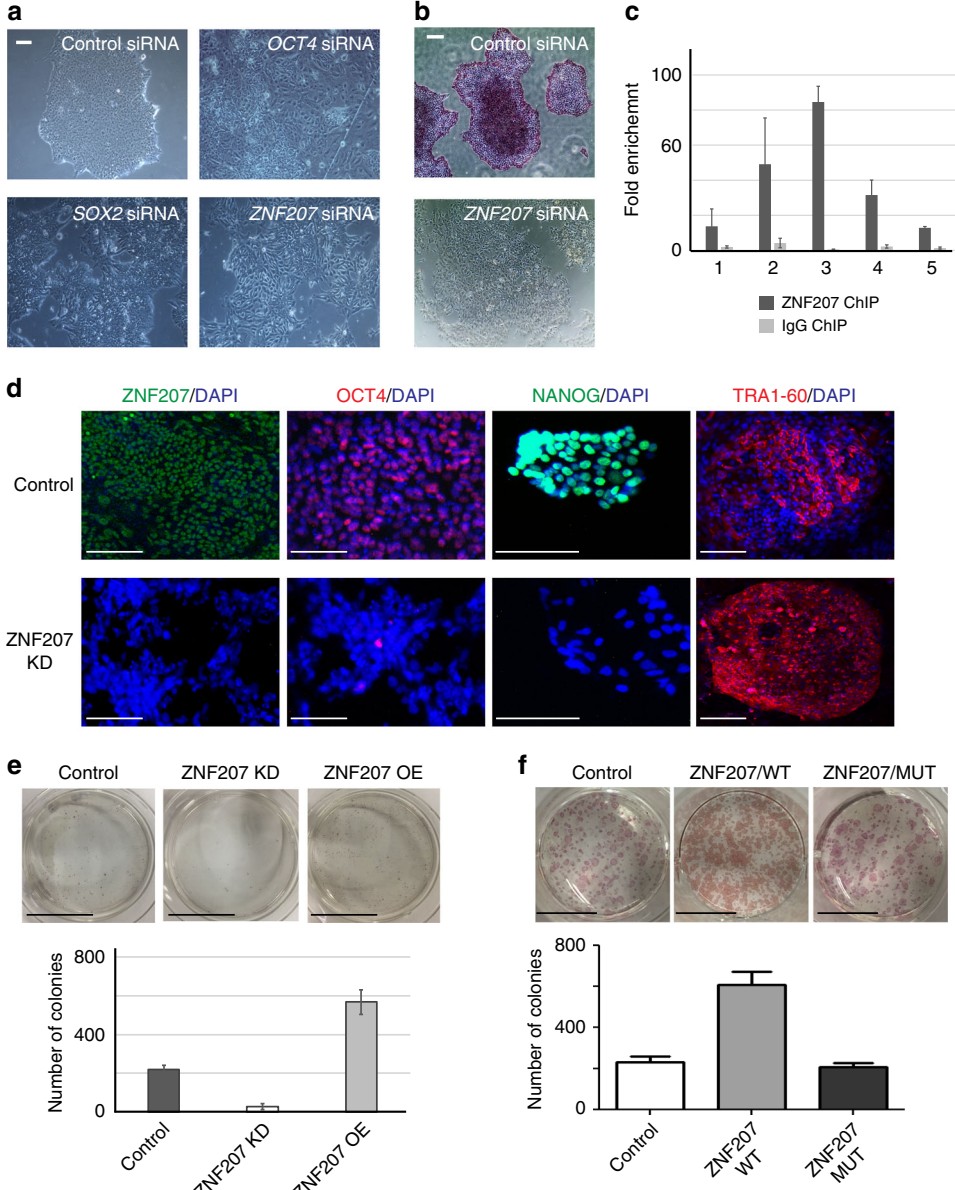

**Fig. 2** ZNF207 is required for hESCs maintenance and enhances reprogramming efficiency through direct regulation of *OCT4*. **a** Bright field view of cell morphology after knockdown. Scale bars represent 50 μm. **b** Alkaline phosphatase staining of cells after knockdown. Scale bars represent 50 μm. **c** Validation of the binding of ZNF207 by ChIP. A ChIP assay was performed using anti-ZNF207 antibody to detect enriched fragments in hESCs. IgG ChIP is served as control. The locations of the amplified products are labeled as Fig. 1b. **d** Immunofluorescence staining of control and ZNF207 KD hESCs. Scale bars represent 50 μm. **e** Overexpression of ZNF207 (ZNF207 OE) enhanced reprogramming efficiency, while knockdown of ZNF207 (ZNF207 KD) reduced the efficiency. Top: Colony formation shown by AP staining. Bottom: Counts of colonies. Scale bars represent 5 cm. **f** Deletion of ZNF207 DNA-binding domain abolished its effect on reprogramming. Top: Colony formation shown by AP staining. Bottom: Counts of colonies. Scale bars represent 11 mm. ZNF207 WT: wild-type ZNF207 protein; ZNF207 MUT: ZNF207 protein without DNA binding domain. Data are presented as the mean ± SEM and are derived from three independent experiments

(iPSCs). Strikingly, we found that KD of ZNF207 during the process of reprogramming greatly reduced the efficiency of reprogramming by about tenfold, while overexpression of ZNF207 enhanced the efficiency by about threefold (Fig. 2e). Note that the change in reprogramming efficiency does not appear to be directly linked to alteration of cell proliferation; the KD of ZNF207 resulted in a decrease in the cell proliferation rate to approximately 75% of that of the control, whereas, OE of ZNF207 had no obvious effects on cell proliferation (Supplementary Fig. 2d). We observed higher expression of endogenous *OCT4* in ZNF207 OE cells and lower expression of endogenous

*OCT4* in ZNF207 KD cells, which correlates with the reprogramming efficiency in each condition (Supplementary Fig. 2e). To better understand the molecular mechanism of ZNF207 in promoting the reprogramming efficiency, we mutated ZNF207 protein by deleting its DNA binding domain (ZNF207/MUT) and introduced it to the reprogramming process, in comparison to the wild-type ZNF207 protein OE (ZNF207/WT) and no ZNF207 OE (Control) conditions. We found that mutated ZNF207 is not able to promote increased reprogramming efficiency (Fig. 2f). This result suggests that ZNF207 promotes reprogramming by regulation of gene transcription through its DNA binding

domain. Taken together, these results indicate that ZNF207 is essential for the maintenance of hESCs and reprogramming of human pluripotent stem cells from somatic cells.

Recently, it has been demonstrated that there are at least two distinct pluripotent states, termed the primed and naive states. hESCs and human-induced pluripotent stem cells (hiPSCs) are most similar to the post-implantation epiblast, whereas, naive-state human pluripotent stem cells (hPSCs) are more comparable to the preimplantation epiblast[24]. To determine whether ZNF207 functions in naive-state hPSCs as well as primed, we performed knockdown analysis in naive hPSCs. The naive human pluripotent stem cells were derived and maintained based on the protocol of Takashima et al.[25] We observed that depletion of ZNF207 in naive-state hPSCs resulted in disruption of colony forming morphology (Supplementary Fig. 3a), loss of alkaline phosphatase staining (Supplementary Fig. 3b), reduced expression of pluripotency genes, and upregulation of lineage marker genes (Supplementary Fig. 3c). These results suggested that ZNF207 may be a critical regulator in human embryo development; thus, we examined whether ZNF207 is expressed in preimplantation human embryos. We observed that ZNF207 is activated at the 8-cell stage, with timing similar to many other factors including *NANOG* (Supplementary Fig. 3d). The expression of *ZNF207*, however, was significantly higher than *NANOG* across all the stages of preimplantation development (Supplementary Fig. 3e)[26,27].

**Identification of direct ZNF207 target genes in hESCs**. To investigate the global regulatory roles of ZNF207 in hESCs, RNA-Seq was used to compare the transcriptomes in ZNF207 KD cells and control hESCs. A total of 1287 genes were found to be downregulated and 785 genes were upregulated by at least two-fold in ZNF207 KD cells compared to control (Fig. 3a). To establish which of the genes, that were differentially expressed between KD and control cells, are directly regulated by ZNF207, ChIP-Seq was used to profile the genome-wide binding sites of ZNF207 in hESCs. Approximately 8000 ZNF207 binding sites were identified across the genome of hESCs, with the strongest binding clustered around the transcription start site (TSS) (Fig. 3b). Note that a previously unknown motif of statistical significance for ZNF207 was identified and direct binding of ZNF207 protein to this DNA sequence was confirmed by electrophoretic mobility shift assays (EMSA). Mutation of specific conserved nucleic acids in the motif abolished the binding of the ZNF207 protein to the DNA sequences (Fig. 3c, d).

ChIP-Seq confirmed that ZNF207 binds to *OCT4* PE, as well as DE, and it co-localizes with master TFs, OCT4, SOX2, and NANOG, at this locus (Fig. 3e). Its direct regulation of transcription through the *OCT4* PE was confirmed by luciferase reporter assays (Supplementary Fig. 4a). Furthermore, the effects of ZNF207 KD were rescued by overexpression of OCT4 (Supplementary Fig. 4b, c), suggesting that OCT4 is a major downstream target of ZNF207 in hESCs. Taken together, these data indicate that OCT4 is one of the direct and major functional targets of ZNF207 in hESCs and in the process of reprogramming from somatic cells to pluripotent cells.

In addition to OCT4, we were interested in identification of downstream targets of ZNF207 in the whole genome of hESCs. We defined that genes that are bound by ZNF207 and differentially expressed in ZNF207 KD cells as direct ZNF207 targets in hESCs. Thus, 177 direct target genes that were upregulated and 357 genes that were downregulated in response to ZNF207 KD were identified (Fig. 3f). The GO (Gene Ontology) terms of ZNF207 negatively correlated genes (upregulated in ZNF207 KD cells) were related to cellular processes such as

programmed cell death, cardiovascular development, angiogenesis, and cholesterol biosynthesis (Fig. 3g, top panel), suggesting the possibility that ZNF207 may play a role in repression of apoptosis and mesoderm development. GO terms of ZNF207 positively correlated genes (downregulated in ZNF207 KD cells) were related to cellular process such as cell proliferation, neurogenesis, and beta-catenin–TCF complex assembly (Fig. 3g, bottom panel), suggesting that ZNF207 may function in cell cycle regulation, ectoderm development, and stem cell signaling pathways.

**ZNF207 is a part of the OCT4/SOX2 core pluripotency network**. To probe its role in pluripotency, we analyzed the in vivo sequence specificity of ZNF207. Interestingly, the OCT4/SOX2 consensus motif was enriched as the most significant motif with an E-value of 5.3E − 059 (Fig. 4a) and we observed overlap of OCT4, SOX2, and ZNF207 binding events (Fig. 4b). This reflects the frequent co-localization of ZNF207 with OCT4/SOX2 across the genome of hESCs and suggests that ZNF207 could be one of the interaction partners of the OCT4/SOX2 core pluripotency network. To test potential protein–protein interactions, we performed co-immunoprecipitation (Co-IP) experiment using hESC cell extracts. We detected ZNF207 protein in OCT4 and SOX2 pull-down and vice versa (Fig. 4c), indicating that ZNF207 forms a complex with OCT4/SOX2 in hESCs through protein–protein interactions. In addition to the OCT4/SOX2 motif, we identified known motifs for neuronal TFs, such as ZIC1/ZIC2, SOX10/SOX3/SOX11, TCF3, and ASCL1 (Supplementary Fig. 4d), suggesting that ZNF207 may play a role in neuronal development upon differentiation.

We then analyzed genes that were co-bound by ZNF207, OCT4, and SOX2 (Fig. 4d) and observed that these genes are enriched in the GO terms of embryo development, ectoderm development, and nerve system development (Fig. 4e). For example, these genes include critical pluripotency marker genes, such as *OCT4*, *SOX2*, *SALL4*, *NANOG*, and *ZNF207* itself (Fig. 4f), and important neuroectoderm developmental genes such as *NES*, *NEUROG3*, *NCAM1*, and *GLI2* (Fig. 4g). Taken together, our data strongly suggested that ZNF207 is part of the OCT4/SOX2 core pluripotency complex and they collaboratively target the pluripotency and ectoderm developmental genes in hESCs.

**ZNF207 promotes self-renewal of hESCs**. We observed co-occurrence of ZNF207 with enhancer markers, P300 and H3K27ac in hESCs (Fig. 5a). Apart from "stem cell maintenance genes" that are also co-bound by OCT4/SOX2 (Supplementary Fig. 5a), genes co-marked by ZNF207/P300/H3K27ac largely belong to categories related to cell cycle processes (Fig. 5b). Individual genes, *CDK1*, *BUB1*, *TERF1*, and *TRIAP1*, for examples, are heavily bound by ZNF207 and enhancer markers, P300 and H3K27ac (Fig. 5c). KD of ZNF207 led to differential expression of a few genes involved in hESCs proliferation (Supplementary Fig. 5b). We found that genes involved in senescence, cell cycle checkpoint and cell cycle phase transition were upregulated, and genes involved in regulation of cyclin-dependent protein kinase (CDK) activity, telomerase activity and DNA damage response were downregulated in ZNF207 KD cells (Fig. 5d). This is consistent with the findings that pluripotent cells exhibit high CDK and telomerase activity, lack checkpoint regulation and do not appear to undergo senescence. When differentiation occurs or pluripotency is lost, cells become more cell cycle regulated and their DNA repair ability is decreased[28–30]. We also confirmed that the change in expression of RNAs of cell cycle genes were reflected in protein levels (Supplementary Fig. 5c). Further cell cycle analysis demonstrated that reduced levels of

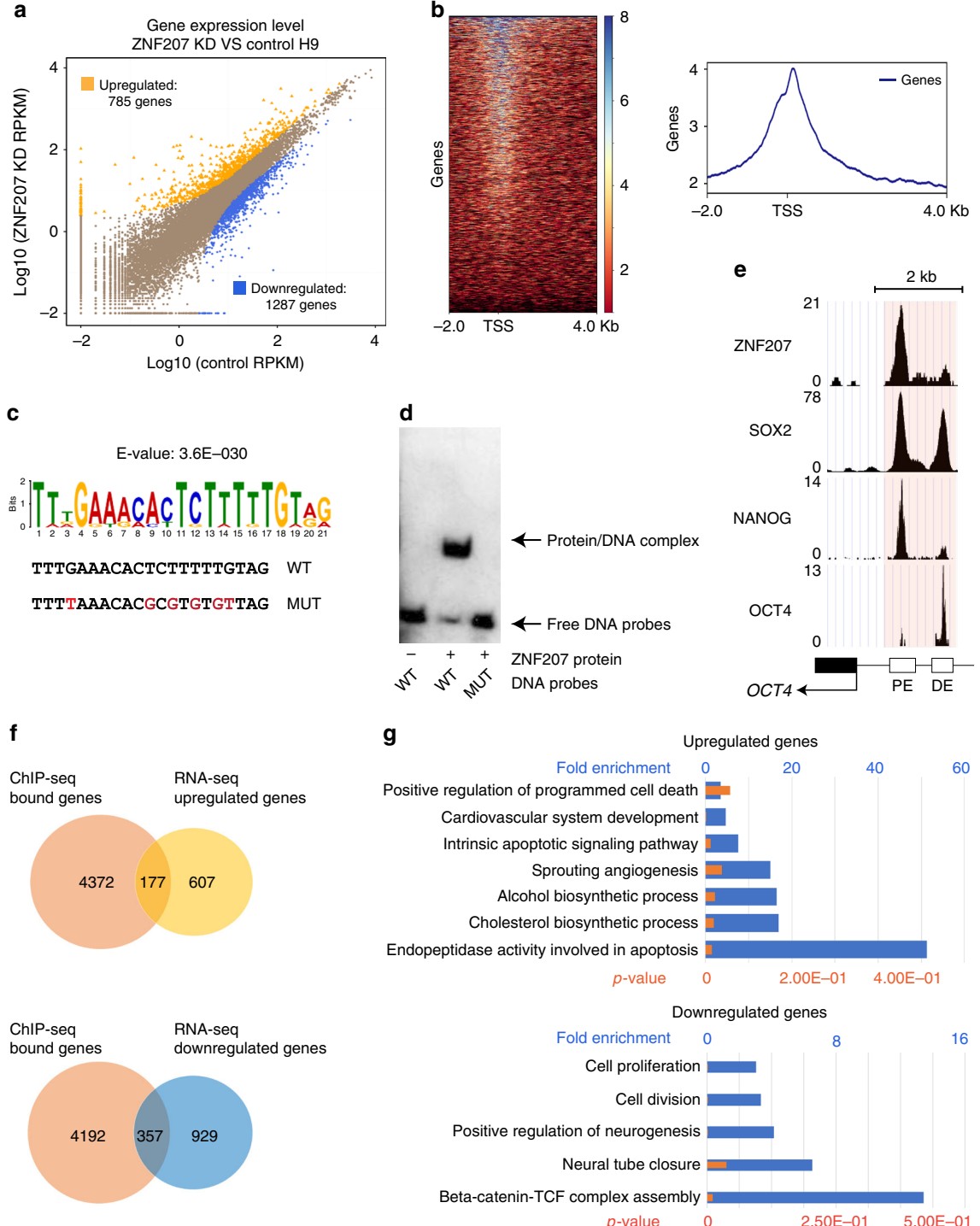

**Fig. 3** High-throughput analysis in hESCs to identify direct targets of ZNF207. **a** Analysis of differential expression on the RNA-Seq data of ZNF207 KD and control hESCs. Genes significantly changed (>twofold change, FDR = 1%) are colored in yellow and blue for upregulated and downregulated, respectively. **b** ZNF207 ChIP-Seq analysis in hESCs. Left: Heatmap depicts ZNF207 ChIP-Seq signals at TSSs. Right: Composite plot shows ZNF207 ChIP-Seq signal is enriched at transcription start sites (TSSs). The gradient blue-to-red color indicates high-to-low counts in the corresponding region. **c** Enriched motif from de novo motif search of sequences under ZNF207 peaks. Wild-type (WT) sequence and mutant (MUT) sequence are shown in the bottom. Mutated nucleic acids are labeled by red fond color. **d** EMSA were performed to detect interaction of ZNF207 protein with DNA sequences. **e** ChIP-Seq tracks show co-localization of ZNF207, SOX2, NANOG, and OCT4 at the proximal enhancer of *OCT4* gene. *OCT4* gene, Proximal enhancer (PE) and distal enhancer (DE) are indicated by the boxes in the bottom. The scale bar indicates the size of the chromosome. The light pink boxes highlight the co-bound regions. **f** Venn diagrams show overlaps of ChIP-Seq bound genes with differentially expressed genes identified from RNA-Seq. **g** Gene ontology analysis of genes that are directed regulated by ZNF207. Blue bars represent fold enrichment and orange bars represent *p*-value. *p*-value is calculated using hypergeometric distribution

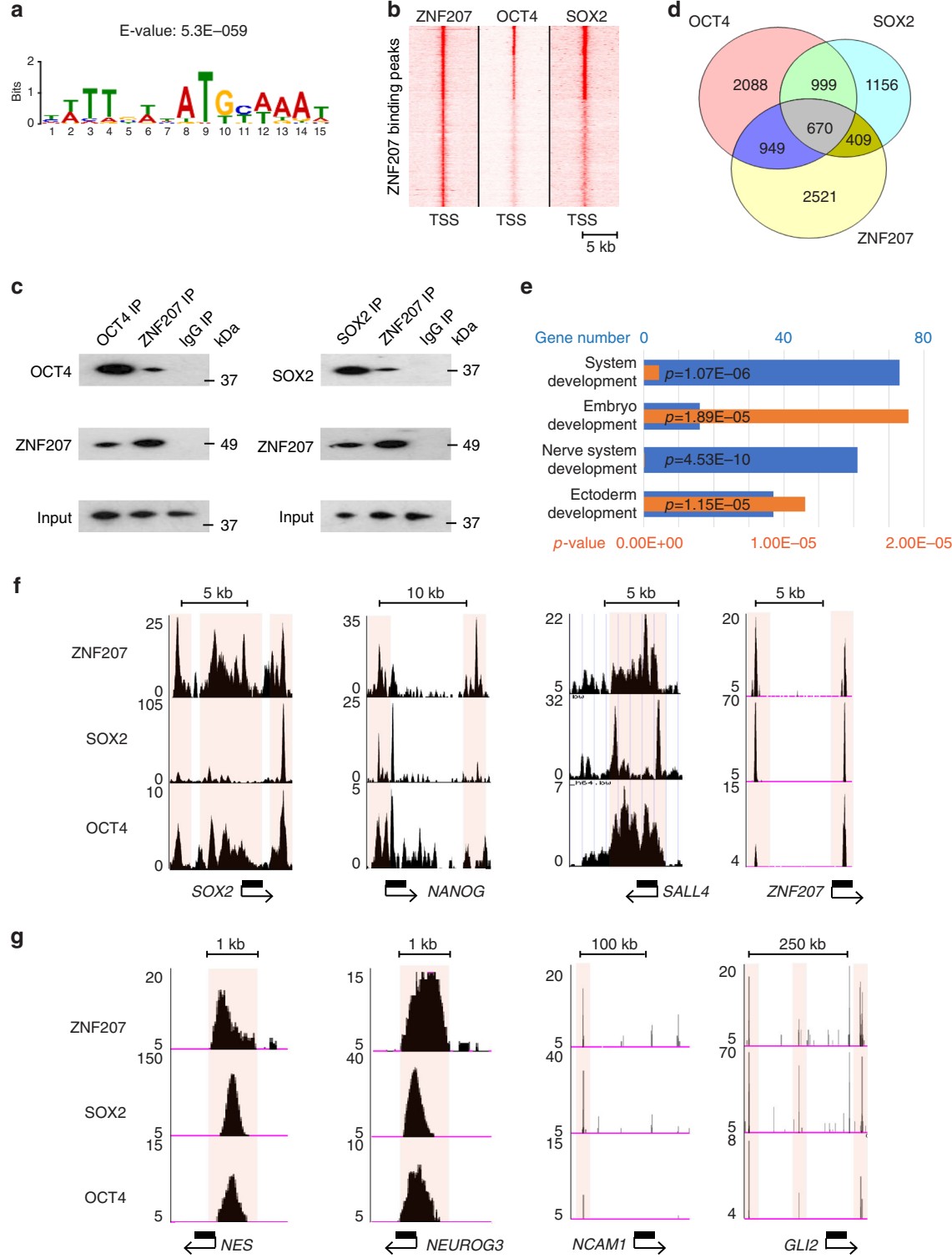

ZNF207 inhibits proliferation of hESCs (Supplementary Fig. 5d) by decreasing the percentage of cells in S and G2/M phase and increasing the percentage of cells in G0/G1 phase (Fig. 5e). These findings suggest that ZNF207 is required for cell cycle progression of hESCs. Reduced cell cycle progression and the lengthening of G0/G1 phase may provide a window for differentiation signals.

More interestingly, we found that *FGF2* (*bFGF*) could be one of the direct targets of ZNF207: ZNF207 binds to the promoter and enhancer regions of *FGF2* gene (Fig. 5f). The mRNA and protein

expression levels of bFGF were correlated with the levels of ZNF207 (Fig. 5g, h). It has been shown that bFGF signaling is of central importance to hESCs self-renewal[31–34]. Thus, we used clonogenicity assays to test whether ZNF207 OE could prolong self-renewal of hESCs in the absence or low concentrations of bFGF. The data indicated that control hESCs failed to survive in 20 ng/ml bFGF culture media, whereas, ZNF207 OE hESCs could proliferate as single cells and form AP-staining positive colonies (Fig. 5i). These results suggest that ZNF207 OE promotes the

**Fig. 4** ZNF207 co-localizes with master transcription factors in hESCs to regulate pluripotency and neuronal gene expression. **a** Enriched motifs from de novo motif search of sequences under ZNF207 peaks. Note the identification of consensus OCT4/SOX2 binding but also other known transcription factor binding motifs. Statistical significance (E-values) is indicated below the motif logo. **b** Heatmap depicts ZNF207, OCT4 (GSM1124067), and SOX2 (GSM1701825) ChIP-Seq signals at TSSs. **c** Protein interactions between ZNF207 with OCT4 and SOX2. Co-IP using cell extracts from hESCs was performed using anti-OCT4 and anti-SOX2 antibody. Western blotting was carried out with anti-ZNF207 antibody. Control IP was performed using anti-IgG antibody. Reverse co-IP was performed using anti-ZNF207 antibody, and western blotting was then performed with anti-OCT4 or anti-SOX2 antibody. Input was shown as loading control. **d** Venn diagrams show overlaps of genes bound by OCT4, SOX2, and ZNF207 in hESCs. **e** Gene ontology analysis of genes that are directed regulated by ZNF207. Blue bars represent the number of genes and orange bars represent *p*-value. *p*-value is calculated using hypergeometric distribution. **f** ChIP-Seq tracks show co-localization of ZNF207, SOX2, and OCT4 at the regulatory sequences of pluripotency genes. The scale bars indicate the size of the chromosome. The light pink boxes highlight the co-bound regions. **g** ChIP-Seq tracks show co-localization of ZNF207, SOX2, and OCT4 at the regulatory sequences of neuronal genes. The scale bars indicate the size of the chromosome. The light pink boxes highlight the co-bound regions

production of bFGF from the cells, prolonging self-renewal of hESCs in low concentrations of bFGF.

**ZNF207 directs specification to ectoderm**. Previous studies suggested that the core pluripotent TFs, OCT4, SOX2, and NANOG may not only play important roles in the induction and maintenance of pluripotency, but may also function as lineage specifiers, regulating the differentiation of ESCs to specific lineages[35–37]. Although high-throughput and loss-of-function studies shed light on ZNF207 as a part of the core pluripotency network that maintains the identity of hESCs, its role in regulating differentiation was not clear. To investigate the differentiation potential of ZNF207 KD hESCs, cells were differentiated in vitro into the three germ layers: ectoderm, mesoderm, and endoderm. ZNF207 siRNAs were transfected into the hESCs every 2 days to ensure that the levels of ZNF207 were constantly reduced during differentiation (Supplementary Fig. 6a). Results of immunostaining revealed higher expression levels of endodermal (SOX17/FOXA2) and mesodermal (NCAM) marker proteins in ZNF207 KD cells and reduced expression of ectodermal marker (NES) proteins compared to control H9 cells (Fig. 6a). To quantify, we counted the total number and the percentage of cells that stained positive for each lineage and found that ZNF207 KD cells were able to differentiate more efficiently to endodermal and mesodermal cells, and less so to ectodermal cells (Fig. 6b). These results indicated that KD of ZNF207 in hESCs disrupts the balance of pluripotency and differentiation (Fig. 6c).

Notably, we observed that *OTX2*, an early ectodermal lineage marker gene, is bound by ZNF207 (Supplementary Fig. 6b). In the process of ectoderm differentiation from hESCs, *OTX2* expression is induced to levels that facilitate further differentiation. In ZNF207 KD cells, *OTX2* expression cannot be induced efficiently and is at much lower level compared to the control. However, its expression is restored by a rescue experiment with ZNF207 OE (Supplementary Fig. 6c). These data indicate that *OTX2* is a direct target of ZNF207. To address the possibility that reduced expression of *OTX2* might be linked to the deficiency in ectoderm differentiation, we overexpressed OTX2 in ZNF207 KD hESCs to the comparable level of control cells (Supplementary Fig. 6d) to test whether this could rescue differentiation to ectoderm. Indeed, overexpression of OTX2 restored the ability of ZNF207 KD cells to differentiate to ectoderm as detected via re-expression of early ectoderm markers, NES and SOX2 (Fig. 6d). We also observed upregulation of *FOXG1* and *NGN2*, which are early neuroectoderm markers that work downstream of *OTX2* (Supplementary Fig. 6d). Quantification analysis showed increased number of ectoderm cells under rescue conditions (ZNF207 KD, OTX2 OE) compared to ZNF207 KD (Fig. 6e). Thus, ZNF207 is part of the core pluripotency network and forms complexes with OCT4/SOX2/P300. It safeguards pluripotency by directly controlling key pluripotency genes and maintaining cell proliferation and self-renewal by activating cell cycle genes. In addition, ZNF207 ensures ectoderm differentiation potential by maintaining expression of early ectoderm genes, such as *OTX2*. KD of ZNF207 causes reduced expression of pluripotency genes, compromises stem cell self-renewal, and blocks ectoderm differentiation (Fig. 6f).

**ZNF207 switches isoforms during cell differentiation**. ZNF207 was originally discovered as a kinetochore-binding protein that promotes spindle assembly during mitosis in HeLa cells and Glioblastoma multiforme stem cells (GSCs)[38–40]. Whether ZNF207 also controls spindle assembly in hESCs is still unknown. To investigate, we knocked down ZNF207 and assessed whether it causes defects in spindle formation as reported in cancer cells (Supplementary Fig. 7a, b). Kinetochores were stained with CREST (centromere) antibodies, microtubules were stained with an anti-α-tubulin antibody, and chromosomes were stained with DAPI. To our surprise, we did not observe much misalignment in mitotic cells in both control and ZNF207 KD hESCs (Fig. 7a). We also quantified the percentage of misaligned chromosomes and found only 0–1% of the cells were misaligned and there was no significant difference between control and KD cells (Fig. 7b).

The data above provided substantial evidence that ZNF207 functions as a critical TF in hESCs to regulate downstream pluripotency and developmental gene expression; a role in regulating mitotic chromosome alignment was not observed. Previous studies indicated that ZNF207 has three isoforms (isoform A, B, and C). Isoform C is the longest isoform; isoform A lacks exon 6, and isoform B lacks exon 9 (Fig. 7c). ZNF207 mRNA is the target of a splicing factor SFRS11 and during somatic reprogramming, ZNF207 changes from isoform B to A and C[41]. Thus, we investigated whether different functions of ZNF207 are linked to the presence of various isoforms in different cell types. We note that we used isoform C for all the ZNF207 protein experiments, including EMSA (Fig. 3e) and overexpression (Figs. 2e, f and 5h, i) analysis. We initiated neuronal differentiation of hESCs and found that ZNF207 gradually switches its isoform from A and C to B (Fig. 7d). We also found that isoform C is dominant in pluripotent stem cells and isoform B is present more in terminally differentiated cells and cancer cells (Supplementary Fig. 7c). The RNA-Seq data showed that the exon 6 and 9 skipping is rare, suggesting that isoform C is the dominant isoform in hESCs (Supplementary Fig. 7d). To provide more functional validation of this isoform change, we assessed rescue of ZNF207 KD via overexpression of three ZNF207 isoforms respectively. Only forced expression of isoform C was able to restore the ZNF207 KD phenotype in hESCs (Fig. 7e, Supplementary Fig. 7e); overexpression of isoform A and B had no rescue effect. Gene expression analysis further demonstrated that isoform C rescued the differentiation by reactivating the expression of downstream targets, such as *SALL4* and *OTX2*, through its transcription factor activity (Fig. 7f). These results

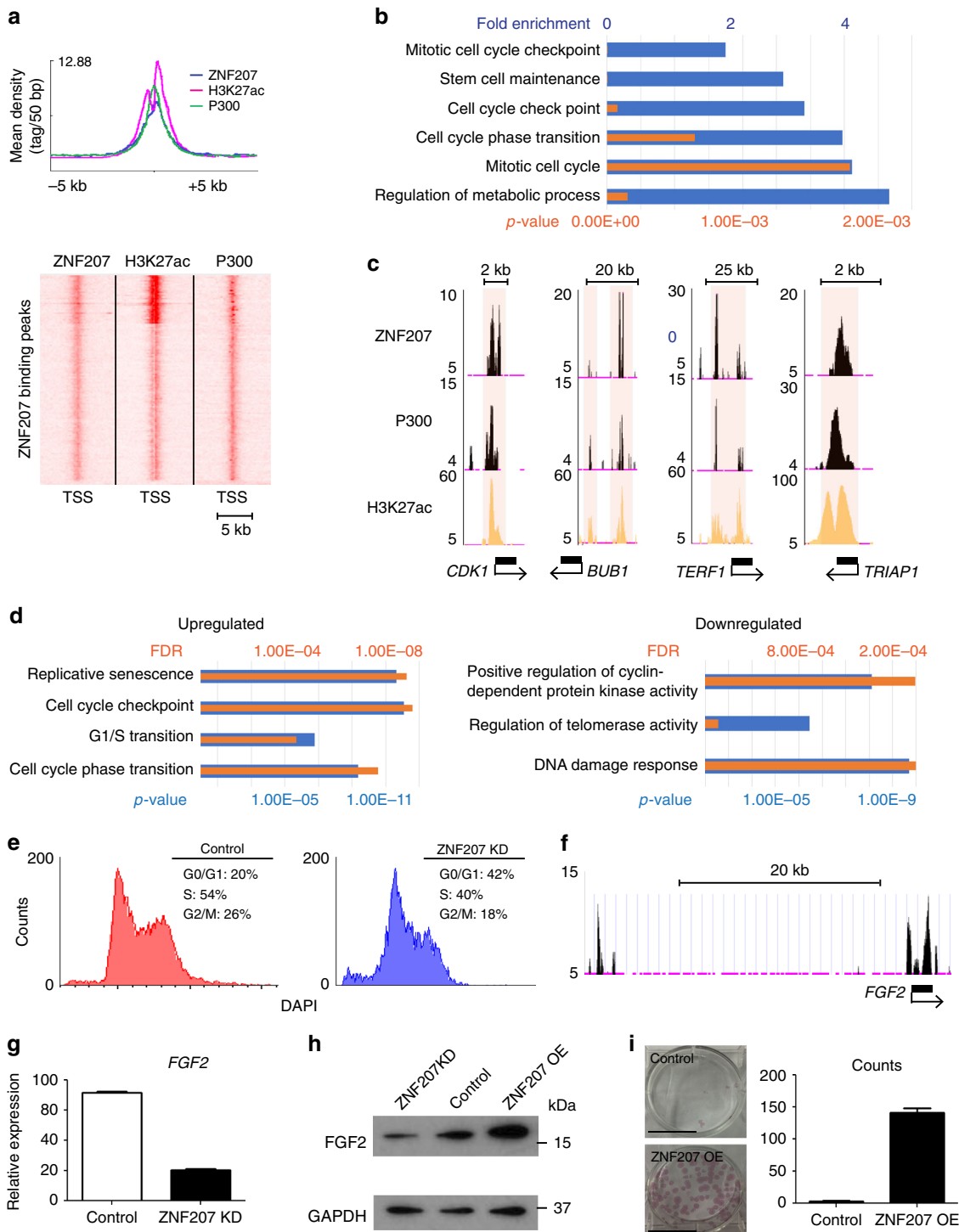

**Fig. 5** ZNF207 co-localizes with enhancer markers to promote cell cycle of hESCs. **a** Top: Composite plot shows ZNF207, p300 (GSM1003513) and H3K27ac (GSM733718) ChIP-Seq signals are enriched and overlapped at TSSs. Bottom: Heatmap illustrates genome-wide association of ZNF207 with p300 and H3K27ac-binding sites. **b** Gene ontology analysis of genes that are co-bound by ZNF207, p300 and H3K27ac. Blue bars represent fold enrichment and orange bars represent *p*-value. *p*-value is calculated using hypergeometric distribution. **c** ChIP-Seq tracks show co-localization of ZNF207, P300, and H3K27ac at the regulatory sequences of cell cycle genes. The scale bars indicate the size of the chromosome. The light pink boxes highlight the co-bound regions. **d** Gene ontology analysis of cell cycle genes that are differentially expressed in ZNF207 KD cells. Blue bars represent *p*-values and orange bars represent FDR (false discovery rate). *p*-value is calculated using hypergeometric distribution. **e** Cell cycle progressions determined by flow cytometry. Histogram plot of flow cytometry analysis of control cells (red) and ZNF207 KD cells (blue). The percentage of cells that are in each phase of mitosis is shown. **f** ChIP-Seq tracks show binding of ZNF207 at promoter and enhancer regions of *FGF2* gene. **g** The expression of *FGF2* in control and ZNF207 KD cells. **h** Western blot to detect the protein level of FGF2 in KD, control, and OE cells. GAPDH is shown as the loading control. **i** Clonogenic assay to show the self-renewal ability of control and ZNF207 OE cells in E7 media with 20 ng/ml bFGF. Scale bars represent 17.5 mm. The counts of colonies are shown on the right. Data are presented as the mean ± SEM and are derived from three independent experiments

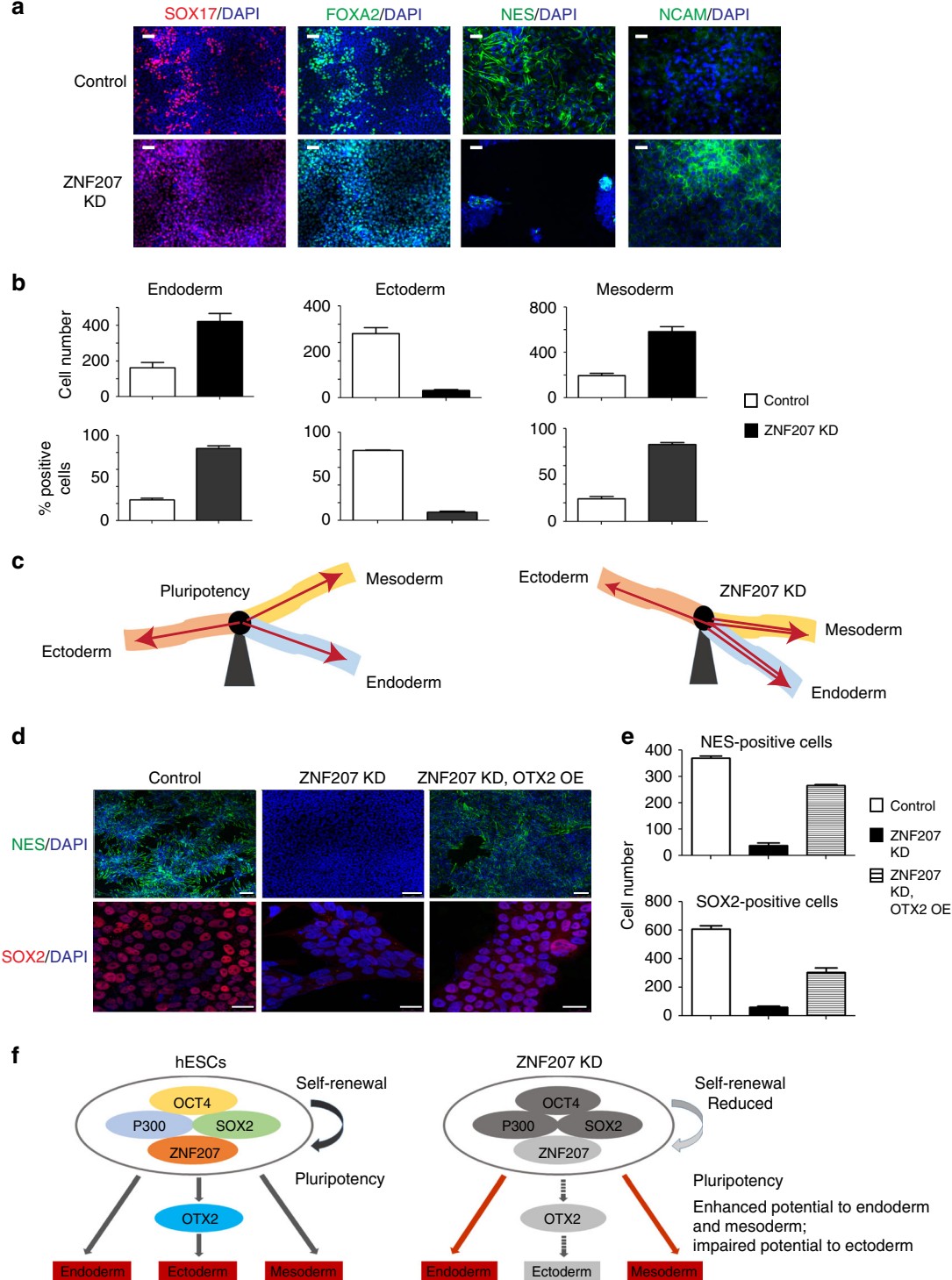

**Fig. 6** ZNF207 works upstream of OTX2 in hESCs to govern specification to ectoderm. **a** Immunofluorescence staining of control and ZNF207 KD hESCs for lineage markers. Scale bars represent 50 μm. **b** Counts and percentage of cells that were stained positive for each lineage. **c** KD of ZNF207 disrupted pluripotency of hESCs. HESCs have equal potential to differentiate into any of the three lineages, while KD of ZNF207 tilts the potential to endoderm and mesoderm. The potential to differentiate into ectoderm is significantly impaired by KD of ZNF207. **d** Immunofluorescence staining of control, ZNF207 KD and ZNF207 KD, OTX2 OE hESCs for ectodermal proteins. Top: Scale bars represent 100 μm; Bottom: Scale bars represent 25 μm. **e** Counts of cells that were stained positive for ectodermal proteins. **f** ZNF207-centered transcriptional network to regulate self-renewal and pluripotency in hESCs. ZNF207 works with master transcription factors and co-factors, such as OCT4, SOX2, and P300, to form the core transcriptional network in hESCs. They bind together at pluripotency, development, and cell cycle-related genes to govern self-renewal and pluripotency. OTX2 is targeted and regulated by ZNF207 to ensure the developmental potential towards ectoderm. Once ZNF207 is knocked down, the expression of OCT4, SOX2, and P300 is reduced. The level of cell cycle-related genes has been changed, which leads to reduced ability of self-renewal. The expression of *OTX2*, which is downstream of ZNF207, is also downregulated, which impedes the differentiation potential towards ectoderm. In the meantime, the potential to endoderm and mesoderm is enhanced. Data are presented as the mean ± SEM and are derived from three independent experiments

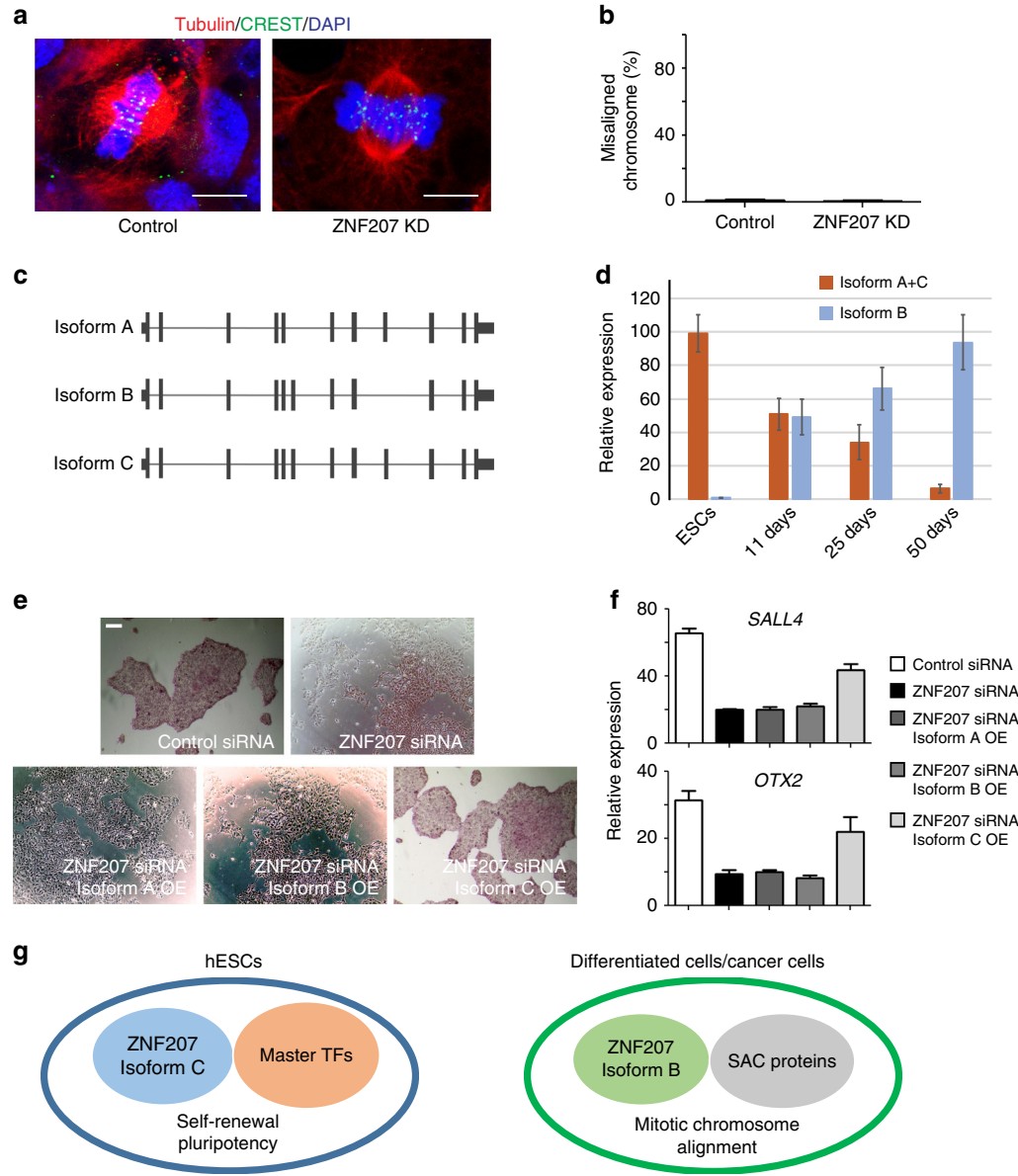

**Fig. 7** ZNF207 switches isoforms during cell differentiation. **a** Confocal images of mitotic cells with aligned chromosomes. ~100 mitotic cells were measured for each experiment and condition. Centromeres were detected by the CREST serum. Scale bar, 5 μm. **b** Counts of mitotic cells with misaligned chromosomes in control and ZNF207 KD cells. **c** Schematic representation of the three alternative splice isoforms of ZNF207. **d** RT-qPCR analysis to test the expression level of different isoforms during induced neuronal differentiation from hESCs. **e** Alkaline phosphatase staining of cells in different conditions. Scale bars represent 50 μm. **f** RT-qPCR analysis to test the expression level of *SALL4* and *OTX2*. Data are presented as the mean ± SEM and are derived from three independent experiments. **g** The model of ZNF207 isoform change in hESCs and differentiated/cancer cells. Data are presented as the mean ± SEM and are derived from three independent experiments

indicated that ZNF207 switches isoforms in different cell types and individual isoform play distinct roles in the cell. Isoform C of ZNF207 is dominantly present in hESCs and is the functional isoform of ZNF207 that interacts with master TFs and co-factors to function as a TF to control the transcription of key genes in order to maintain self-renewal and pluripotency of hESCs. In differentiated cells and cancer cells, isoform B of ZNF207 interacts with spindle assembly checkpoint (SAC) proteins to regulate mitotic chromosome alignment (Fig. 7g).

## Discussion

The OCT4 TF is required to induce and maintain self-renewal and pluripotency in pluripotent stem cells[1,42]. In this study,

endogenous regulation of *OCT4* in hESCs was probed through locus-specific proteomics and ZNF207 was identified as a critical regulator for self-renewal and pluripotency and a key driver of differentiation of hESCs to ectoderm. On the one hand, ZNF207 works with master TFs as a guardian of pluripotency and self-renewal in hESCs controlling the expression of pluripotency genes such as *OCT4*, *P300*, *SALL4*, and *PRDM14*, as well as self-renewal genes such as *CDK1*, *BUB1*, and *TERF*. Dysregulation of any of these genes is known to induce loss of hESC pluripotency and self-renewal. Importantly, data suggests that ZNF207 functions to drive initial differentiation, especially the differentiation potential towards ectoderm. As indicated in the Waddington's canal[43], wild-type hESCs have equal potential to differentiate into any of the three primary germ layers if provided appropriate

conditions. However, KD of ZNF207 tilts the potential of differentiation towards endoderm and mesoderm; Their potential to become ectoderm has been significantly impaired (Fig. 6c). It has been proposed that pluripotency TFs may not work as inhibitory TFs that cooperatively block differentiation to all lineages but instead may modulate lineage specifiers that direct commitment to different lineages[35–37]. For example, Nanog enhances the ability to differentiate into definitive endoderm[44] and SALL4 is required for specification of trophoblasts[45]. Similarly, Sox2 and Otx2 are implicated in ectoderm specification[46]. We add ZNF207 to the membership of the pluripotency/ectoderm specification factor cascade. ZNF207 promotes ectoderm differentiation by activating early neuronal genes, including *OTX2*, a gene required for ESC-derived neural precursors to differentiate into telencephalon and mesencephalon[47,48]. The ability of ZNF207 to activate and maintain *OTX2* expression while promoting hESC self-renewal and pluripotency is indicative of an early driver of hESC differentiation to ectoderm.

These studies report that in hESCs, isoform A and C of ZNF207 are abundant and isoform C functions as a transcription factor to maintain the core transcriptional network. Once hESCs differentiate, ZNF207 switches from isoform A and C to isoform B. Our data, together with previous reports, suggests that this change of isoform switches the function of ZNF207 from a critical transcription regulator to a SAC regulator during mitosis. We hypothesize that switching isoforms may be a common underlying mechanism for TFs to play distinct roles in various cell types during development. Notably, during differentiation, 489 genes in hESCs may change isoforms including 33 which are classified as TFs[41].

In conclusion, these studies demonstrated that a locus-specific proteomics approach is able to identify known *OCT4* regulators, such as OCT4, SOX2, SALL4, and HDAC2, and also identify previously unknown proteins that regulate *OCT4* expression. Findings illuminate specific aspects of transcriptional regulation in pluripotent stem cells while expanding our understanding of the intrinsic transcriptional network that controls identity of hESCs and promotes differentiation to one of three primary germ layers in human development, the ectoderm. Furthermore, given the expression of ZNF207 in human embryos, it is of interest to investigate the roles of ZNF207 in early human embryo development at pre-implantation and/or post-implantation stages.

## Methods

**Cell culture.** Human ES cell-line (H9, WiCell) was cultured feeder-free on matrigel matrix (BD Biosciences) and maintained in mTeSR media (STEMCELL Technologies)[49]. In the bFGF clonogenic experiments, hESCs were culture in Essential 6 medium (ThermoFisher; A1516401) with addition of 2 ng/ml TGFβ (R&D systems) and 20 ng/ml bFGF (Pepro Tech). Naive hPSCs were kindly provided by Dr. Sarita P. Panula[50] and cultured in t2i/L + PKCi conditions in a 1:1 mixture of DMEM/F12 and Neurobasal, 0.5x N2-supplement, 0.5x B27-supplement, 1x nonessential amino acids, 2 mM L-Glutamine, 1x Penicillin/Streptomycin (all from ThermoFisher Scientific), 0.1 mM β-mercaptoethanol (Sigma-Aldrich), 1 μM PD0325901, 1 μM CHIR99021, 20 ng/ml human LIF (all from WT-MRC Cambridge Stem Cell Institute), and 2 μM Gö6983 (PKCi; Tocris) on a MEF-layer seeded at a density of $2 \times 10^6$ cells per 6-well plate[25].

**Construction of TALE plasmid.** Modified TALE constructs targeting *OCT4* proximal enhancer were designed and assembled based on the backbone of Daniel Voytas lab golden gate talen assembly 2.0 (Addgene, Kit #1000000024). A control TALE protein targeting green fluorescence protein (GFP) was also designed and assembled. For hESC overexpression, the CMV promoter was replaced with human EF1α promoter. For affinity enrichment of chromatin from the proximal enhancer region of the *OCT4* gene in hESCs, TALE proteins were designed to bind genomic sequences of *OCT4* proximal enhancer (Supplementary Table 1). The TALE protein was designed as a truncation that lacked Fok I nuclease domain. The 3xFLAG tag was added to the N-terminus and maxGFP was added to the C-terminus for visualization. Lentiviral version of modified TALE plasmid was constructed by subcloning the TALE ORF to the lentiviral overexpression vector pLex307 (Addgene #41392).

**Transfection.** Transfection of dTALE construct in hESCs was performed using Fugene 6 (Promega) according to manufacturer's instructions. Puromycin (Sigma) selection was introduced 1 day after transfection at 2.0 μg/ml.

**Virus generation.** Lentiviruses were produced in HEK293T cells (ATCC, VA) by cotransfection with two helper plasmids (pVSVG and pDelta8.9: 12 μg of lentiviral vector DNA and 6 mg of each of the helper plasmid DNA per 75 cm² culture area) using Lipofectamine 2000 (Invitrogen)[51]. Lentiviruses were harvested with the medium 46 h after transfection, pelleted by centrifugation ($49,000 \times g$ for 90 min), resuspended in MEM, aliquoted, and snap-frozen in liquid $N_2$. Only virus preparations with >90% infection efficiency as assessed by EGFP expression or puromycin resistance were used for experiments. The sequence of ZNF207 OE construct is shown in Supplementary Table 4.

**Viral transduction.** Undifferentiated H9 cells were transduced on Matrigel-coated plates with dTALE or plex307-ZNF207 (isoform C) or plex307-OTX2 overexpression vectors along with viral supernatant supplemented with 8 μg/ml polybrene (Sigma-Aldrich). After 6 h, Virus supernatants were washed off, and fresh mTESR1 medium was added. Subsequently, cells were selected with 2 μg/ml puromycin (Invitrogen) for 4 days in mTESR1 media.

**Co-immunoprecipitation.** Transfected cells were lysed in cell lysis buffer (50 mM Tris-HCl pH 8.0, 150 mM NaCl, 1 mM EDTA, 1% NP40, 10% glycerol with protease inhibitor cocktail) for 1 h. Whole-cell extracts were collected and precleared. Beads coated with FLAG antibody (CST #2368), OCT4 antibody (sc-8628, Santa Cruz Biotechnology), and ZNF207 antibody (PA5-30641, ThermoFisher Scientific) were incubated with precleared whole-cell extracts at 4 °C overnight. The beads were washed with cell lysis buffer four times. Finally, the beads were boiled in 2x sample buffer for 10 min. The eluents were analyzed by Western blot.

**Mass spectrometry.** Briefly, $10^{11}$ OCT4 targeting TALE and control GFP targeting TALE transduced hESC cells were crosslinked by 1% formaldehyde, then collected, and lysed by ChIP lysis buffer. Genomic DNA was sonicated to 1 kb. Proteins complexes co-enriching with the TALE-FLAG or proteins non-specifically binding to the anti-FLAG Dynabeads (Invitrogen) were reverse crosslinked by incubation at 65 °C for 6 h. Proteins were purified by acetone precipitation and then resuspended in Tris-HCl buffer. A fraction of the protein samples was then resolved by SDS–PAGE/Silver-staining. The protein sample was concentrated by using a Microcon YM-10 column (Millipore) according to the manufacturer's protocol. Approximately 10 μl of sample was obtained from the filter device. A 1 μl portion of 100 mmol/l DTT (Sigma) was added, and the resulting mixture was incubated at 45 °C for 30 min. A 1 μl portion of trypsin (Sigma) was added afterward at 37 °C. The overnight digestion and reduction reaction was terminated by adding 0.1 μl of TFA to the digest. LC/MS/MS was performed on an 1100 series capillary LC system (Agilent Technologies) coupled to an LTQ-Orbitrap mass spectrometer (Thermo) operating in positive mode and equipped with a nanospray source. All MS/MS spectra were analyzed by using the SEQUEST algorithm, incorporated in Bioworks software, version 3.1 SR1 (Thermo Finnigan). Peptide fragment lists were generated and submitted to the Swiss-Prot database. Two independent experiments were performed. LC/MS/MS was performed by Caprion Biosciences, Inc.

**Chromatin immunoprecipitation and ChIP-Seq.** ChIP assays were carried out as described[52]. Briefly, cells were crosslinked with 1% formaldehyde for 10 min at room temperature, followed by the addition of 0.2 M glycine to inactivate the formaldehyde. Cells were then lysed to obtain chromatin extracts, which were sonicated to obtain DNA fragments with an average size of 300–500 bp. The resulting chromatin extracts were immunoprecipitated by antibodies immobilized on Protein-G beads. The antibody information is shown in Supplementary Table 10. Relative occupancy values (fold enrichment) were calculated by determining the apparent immunoprecipitation efficiency and normalized to the level observed at a control region, which was defined as 1.0. All ChIP experiments were repeated at least three times and data are presented as the mean ± SEM. Primer sequences are available in Supplementary Table 5. For ChIP-Seq, Illumina HiSeq 2 × 60 bp paired-end reads were used for sequencing.

**Motif analysis.** Motif analysis was performed using MEME-ChIP suite with default parameters[53]. The motifs for ZNF207 were searched in TRANSFAC and JASPER databases to find transcription factors with similar consensus sequences[54,55].

**Electrophoretic mobility shift assays.** Recombinant human ZNF207 protein was obtained from Novus Biologicals, LLC (H00007756-P01-10μg). Double-stranded DNA oligonucleotides labeled with biotin at the 5' termini of the sense strands (Integrated DNA Technologies, Inc) were annealed with reverse strands in annealing buffer (10 mM Tris-HCl, pH 8.0, 50 mM NaCl, 1 mM EDTA) and purified with an agarose gel DNA extraction kit (Qiagen). The sense strand sequence is shown in Fig. 3d. EMSA was performed in a 10-μl reaction mixture

containing 10 mM HEPES, pH7.5, 10 mM KCl, 10 mM MgCl₂, 1 mM DTT, 1 mM EDTA, 10% glycerol, 3 ng of biotin-labeled oligonucleotide, 1 μg of poly(dI-dC) (Amersham) and 100 ng recombinant ZNF207 protein (isoform C). Binding reaction mixtures were incubated for 10 min at room temperature and then subjected to electrophoresis on pre-run 5% native PAGE gels in 0.5x TBE buffer. Gels were transferred to Biodyne B nylon membranes (Pierce Biotechnologies) and detected with LightShift Chemiluminescent EMSA kit (Pierce Biotechnologies).

**Knockdown by siRNA**. Three siRNAs against each candidate gene were purchased from Invitrogen^TM Silencer^TM Pre-designed siRNAs. The sequence of siRNA is listed in Supplementary Table 5, 6. siRNA transfection was performed with Lipofectamine RNAiMAX (Invitrogen). In brief, for each well of 12-well plate, 6 pmol of siRNA were mixed with 1 μl of Lipofectamine RNAiMAX in 100ul of Opti-MEM medium (Invitrogen). The mixture was added to 1–3 × 10⁵ cells with 1 ml of cell culture medium without antibiotics. Twenty-four hours after transfection, the cell culture medium was replaced with fresh medium. Four days after transfection, cells were imaged or harvested for RNA extraction.

**RNA extraction, reverse transcription, and quantitative real-time PCR**. Total RNA was extracted using TRIzol Reagent (Invitrogen) and purified with the RNeasy Mini Kit (Qiagen). Reverse transcription was performed using SuperScript II Kit (Invitrogen). Quantitative PCR analyses were performed in real time using an ABI PRISM 7900 sequence detection system and SYBR green master mix (Life Technologies). The primer information is provided in Supplementary Table 8. The levels of the transcripts were normalized against control empty vector transfection. Data are presented as the mean ± SEM and derived from three independent experiments.

**RNA-Seq and differential expression analysis**. Total RNA was extracted using Acruturus PicoPure RNA Isolation Kit (Life Technologies). RNA quality was determined with Bioanalyzer 2100 (Agilent). Sequencing libraries were constructed by SMARTer universal low input RNA Kit (Clonetech) according to the manufacturer's instructions. DNA library samples were submitted to the Stanford Genomics Facility and 100-base paired-end high-throughput sequencing was performed. All sequenced libraries were mapped to the human genome using TopHat and Cufflink[56,57] with default parameter setup. Differential expression was analyzed using StrandNGS (AvadisNGS).

**Gene ontology analysis**. The Gene Ontology (GO) enrichment was done using GREAT analysis, with default parameters[58]. For Stringent analysis, we set-up the gene association rule to associate TSS only in 15 KB distance.

**Immunofluorescence analysis**. Cells were fixed with 4% paraformaldehyde at room temperature for 20 min, permeabilized with 0.3% Triton X-100 in PBS (PBST) for 10 min, and blocked for 45 min at room temperature in PBST containing 5% normal donkey serum (Jackson ImmunoResearch Laboratories). Primary antibodies were diluted in blocking solution and incubated at 4 °C overnight. Appropriate Alexa Fluor 488 or 594-conjugated secondary antibodies (Jackson ImmunoResearch Laboratories) were diluted in PBS containing 0.1% bovine serum albumin (BSA) and incubated at room temperature for 1 h. Finally, 1 μg/ml DAPI was used for nuclear staining. The information of primary and secondary antibodies used in this study are listed in Supplementary Table 9.

**Western blot analysis**. The cells were washed with cold PBS and then scraped from the plate in 1 ml PBS plus 2 × protease inhibitors (Complete Mini, Roche) and transferred immediately to a 1.5 ml tube on ice. The cell suspension was then spun at 5000 r.p.m. in a microcentrifuge for 3 min and the supernatant was discarded. The cell pellet was resuspended with 200 μl RIPA buffer (50 mM Tris, 150 mM NaCl, 0.5% Sodium deoxycholate, 1% NP-40, 0.1% SDS, pH 8) plus 2 × protease inhibitors (Complete Mini, Roche). The cell pellet suspension was pipetted rigorously at least 10 times, then vortexed for 30 s. The suspension was again spun down for 3 min at the same speed. The supernatant was denatured in 4XNuPAGE LDS sample buffer (ThermoFisher) at 95 °C for 5 min, then loaded onto a precast NuPAGE 10% Bis-Tris midi protein gel (Invitrogen). The SDS-PAGE gels were run at 150 volts for 1 h and transferred to a PVDF membrane by iBlot transfer system (Invitrogen). Transferred blots were blocked in 5% non-fat milk for 1 h at room temperature. The blot was subjected to 1 h of primary antibody incubation, followed by two quick rinses and three washes for 5 min in TBST (TBS, pH 7.4 with 0.1% Tween-20). Secondary antibody incubation had the same duration and washes. ECL + (Amersham) was used to detect the HRP signal and the western blot image was collected using Chemidoc XRS (Bio-Rad). Uncropped western blots are shown in Supplementary Fig. 8. The information for specific antibodies are listed in Supplementary Table 9.

**Luciferase assay for enhancer activity**. Enhancer sequences were generated by PCR of human genomic DNA discovered for gene OCT4 and then cloned into

pGL4.28 (Promega) with restricted enzymes KpnI and BmtI (NEB). The minimal promoter pGL4.28 is used as negative control. Luciferase activity was measured using dual-luciferase reporter assay system (Promega) as described in the manufacturer's Manu.

**FACS and flow cytometry**. Cells may be fixed with 4% paraformaldehyde and stained with DAPI at room temperature for 15 min. Then, cells were dissociated in 0.25% trypsin--EDTA (Gibco BRL) at 37 °C for 5 min and collected by centrifugation at 200×g in an Eppendorf 5702R centrifuge. Then the cells were passed through the 70uM strainers (BD Biosciences) to make sure they were digested as single cells before they were subject to the flow cytometry. The analysis was performed using LSRII (Becton Dickinson) and FlowJo software (Tree Star).

**Differentiation of hESCs to three lineages**. hESCs were differentiated using STEMdiff^TM trilineage differentiation kit (#05230, Stem Cell Technology) based on manufacture's protocol.

**Highlights**.

- Genome-wide, unbiased locus-specific proteomics identifies endogenous regulators of OCT4 in human embryonic stem cells (hESCs).
- ZNF207 interacts with master TFs to maintain hESC self-renewal and pluripotency.
- ZNF207 enhances reprogramming efficiency through induction of endogenous OCT4 expression.
- ZNF207 promotes self-renewal of hESCs.
- ZNF207 controls commitment to ectoderm.
- Different isoforms and functions of ZNF207 occur in various cell types.

## Data availability

The ChIP-seq data and RNA-seq data of ZNF207 in hESCs have been deposited in the NCBI Gene Expression Omnibus database (http://www.ncbi.nlm.nih.gov/geo/) under the accession code GSE118638. The mass spectrometry proteomics data have been deposited to the ProteomeXchange Consortium (http://proteomecentral.proteomexchange.org) via the PRIDE partner repository with the data set identifier PXD010963. All other data supporting the findings of this study are available from the corresponding author upon reasonable request.

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

## Acknowledgements

The work was supported by Montana Board of Research and Commercialization Technology #18-51-037. We thank Dr. Sarita P. Panula for providing the naive state stem cells. We thank Dr. Edward Dratz, Dr. Robert Usselman, and Dr. Amander Clark for helpful scientific discussions.

## Author contributions

The study was conceived and designed by F.F., X.N., and R.R.P.; F.F and X.N. performed most experiments (including proteomics, ChIP-seq, RNA-seq, knockdown, immunostaining, protein pull-down, EMSA, FACS, and luciferase reporter assay) and analyzed the data. B.A., J.C., and Z.C. performed immunostaining and counting for analyzing the function of ZNF207 in spindle assembly in hESCs and cancer cells. J.D.-D. and V.S. performed single-cell RNA-seq analysis for preimplantation human embryos. T.B. made viral vector based constructs for initial knockdown screening. The manuscript was written by F.F. and R.R.P. with input from the other authors.

## Additional information

**Competing interests:** The authors declare no competing interests.

