## [Peer Review file · Nature Communications]

Reviewers' comments:

Reviewer #1 (Remarks to the Author):

Despite extensive studies in understanding self-renewal and pluripotency in mouse embryonic stem cells (ESCs), relatively less is known about how self-renewal and pluripotency is regulated in human ESCs. In this study, Fang et al. performed a locus specific proteomics study to identify novel endogenous regulators of proximal enhancer (PE) of OCT4 in hESCs and identified a number of previously unappreciated factors. Then they focused on one candidate, ZNF207 for detailed mechanistic studies with genomic approaches such as ChIP-seq and RNA-seq analyses. They found that ZNF207 interacts with master TFs (OCT4/SOX2) to maintain hESC self-renewal and pluripotency, and enhances somatic cell reprogramming efficiency through induction of endogenous OCT4 expression. They also found that ZNF207 mainly controls cell cycle regulators. Like other master TFs, they discovered that ZNF207 can function as ectoderm lineage driver through direct binding and regulation of neuronal TFs. Finally, they found that the distinct role of ZNF207 during differentiation occurs via the isoform switching.

While manuscript is clearly written and data well presented in supporting for the conclusions drawn, the technique used and some findings are not novel. Many similar locus specific proteomics approaches have been published such as the one in mouse ESCs¹. And ZNF207 isoform switch in pluripotency control has been reported during the reprogramming process². Nonetheless, the study does provide additional insights into our understanding of human ESC pluripotency by integrating ZNF207 into the core pluripotency regulatory network. I have the following critiques that can further enhance the clarity and impact of the current story.

Main critiques:

1. Since the proteomics screen identified ~150 proteins, it would be nice to have a broader functional test of candidates by loss of function studies besides ZNF207 to show the robustness of the approach in identification of novel factors with functional significance in controlling human pluripotency, then narrow down to ZNF207 for a detailed mechanistic dissection. The authors could utilize previously published RNAi screens for the maintenance and exit of pluripotency (From the Huckhui Ng Lab publications) to evaluate how many of their candidates are present in the previous screens. ZNF207 may be present already in those previous screens. Then pick some of them that are not present in previous screens and test them here by shRNA/siRNA experiments.
2. The loss of function of ZNF207 in compromising reprogramming could be due to the defect on somatic cell proliferation during the reprogramming. Does ZNF207 KD affect somatic cell growth? Also, based on the Loh study², the isoform switch in pluripotency has been detailed during the reprogramming process, does the KD in this study abrogate all the isoforms or specific isoform of ZNF207 during the reprogramming? In other words, can isoform specific shRNA be tested in reprogramming?
3. Fig. 4a/b shows the composition of OCT4/SOX2 motif, but the coIP is only validated for OCT4. How about the SOX2-ZNF207 interaction?

Minor points:

1. Line 102: DNaseI should be DNase1
2. Line 280: ZNF07 should be ZNF207
3. Line 288: Figure7c should be Figure 7c with a space.
4. Line 293: Isoform should be isoform.
5. Line 294-297: Figure S6a/b should be Figure S7a/b.
6. I don't understand two places why ref#32 was cited: 1) Line 290; 2) Line 342. The ref#32 has nothing to do what were mentioned there.
7. Fig. 1c, 3e, 4d: it would be nicer if color coded text is used to match the colored bars. For example, in Figure 4d: "Gene number" in blue and "p-value" in orange to match the same color bars.
8. Fig. 4b: the western blots are too grainy. Not sure if Photoshop background reduction or what.

References

- 1 Rafiee, M. R., Girardot, C., Sigismondo, G. & Krijgsveld, J. Expanding the Circuitry of Pluripotency by Selective Isolation of Chromatin-Associated Proteins. *Molecular cell* 64, 624-635, doi:10.1016/j.molcel.2016.09.019 (2016).
- 2 Toh, C. X., Chan, J. W., Chong, Z. S., Wang, H. F., Guo, H. C., Satapathy, S., Ma, D., Goh, G. Y., Khattar, E., Yang, L., Tergaonkar, V., Chang, Y. T., Collins, J. J., Daley, G. Q., Wee, K. B., Farran, C. A., Li, H., Lim, Y. P., Bard, F. A. & Loh, Y. H. RNAi Reveals Phase-Specific Global Regulators of Human Somatic Cell Reprogramming. *Cell Rep* 15, 2597-2607, doi:10.1016/j.celrep.2016.05.049 (2016).

Reviewer #2 (Remarks to the Author):

In the paper "A distinct isoform of ZNF207 controls self-renewal and pluripotency of human embryonic stem cells in conjunction with master transcription factors" the authors set out to understand the regulation of OCT4 in human pluripotent stem cells. For this, they first established a proteomic approach based on the proximal enhancer of OCT4 and isolated proteins that could bind to this enhancer. Through mass spec, the authors identified known and unknown factors that might be involved in the regulation of OCT4. The authors focused on ZNF207 for the remainder of the paper and studied the function of this protein in primed pluripotency. One of the most exciting findings of the paper is that ZNF207 has different isoforms that show vastly different functions. While ZNF207 has been extensively studied in the context of kinetochores and loss of it interferes with spindle assembly in somatic cells, in human embryonic stem cells a different isoform seems to be expressed that does not interfere with spindle assembly but instead affects pluripotency program as a transcription factor. The functions of Znf207/BugZ has been studied in mitosis: ZNF207 is important for the spindle assembly in somatic cells (Toledo et al. *Dev Cell*, 2014). In addition, ZNF207 is involved in splicing in pre-mRNA (Wan et al. *JCB*, 2015).

ZNF207 had been previously shown to affect reprogramming efficiency (see Toh et al, 2016, *Cell Reports*) and it was also shown that ZNF207 has different isoforms that affect reprogramming in different forms. However, the underlying mechanism was still unclear.

The authors here set out to analyze the underlying function of ZNF207 in pluripotent stem cells and find that it might be acting as a transcription factor. While the conclusions are very exciting and of interest for the community, they are not supported by the data presented and in its current form, the manuscript would not warrant publication in *Nature Communications*. In particular, many controls are missing. However, the manuscript can be strengthened by addressing points laid out below.

General Comment:

Overall, I am very disappointed with the quality of parts of the manuscript. For example, the microscopy images shown in figures 2d, 5e, 6a and d are all very low quality and reveal very little detail of the cells (for example, the authors claim that TRA1-60 is lost from the cell membrane and now found in the cells. That is not very clear from the data presented). If possible, the authors should try to find better examples or maybe repeat some of the experiments. In the current form, the images do not show very much.

Major Points:

- The author's main conclusion is that ZNF207 acts as a transcription factor. These data are supported by CHIP-seq experiments and knock down experiments. Both techniques are not able to conclusively show that it is actually a transcription factor in addition to its roles in mitosis. For example, does ZNF207 bind DNA directly? The data shown suggest that it localizes to DNA, but it does not show an

affinity for DNA. In these cases, I would expect either a mutational analysis of ZNF207, identification of the DNA binding domain or a direct readout of DNA-binding affinity through gel shift assays etc. The authors present a potential motif for DNA binding, but that could also be anything, bound by any other TF in the vicinity of OCT4. A mutational analysis with concomitant loss of the binding would be necessary to further corroborate these findings.

- Figure 7: The most interesting finding of the manuscript is that the function of Znf207 in pluripotent stem cells seems to be different from its function in somatic cells. For this, the authors perform immunofluorescence of Tubulin, CREST in combination with DAPI in WT and in Znf207KD cells. In pluripotent cells, the KD did not affect the alignment of the chromosomes and the authors conclude that the function of Znf207 in pluripotent stem cells is different. However, these conclusions have to be supported by data showing that under the conditions that the authors are working, Znf207KD in somatic cells does indeed affect chromosome alignment. The authors used siRNAs that were ordered from a company. To publish such a negative result (no misaligned chromosomes) it is necessary to show that the underlying experimental conditions can support their conclusions.

- Furthermore, a second role described for Znf207 was shown to be its role in splicing and downstream from this R-loop formation. Did the authors consider this possibility and exclude it in this manuscript?

- Figure 1: It is unclear what kind of controls were used in the initial proteomic analysis? Did the authors include a negative region such as a TALEN designed towards an inactive region? How many independent experiments were performed and how much overlap existed between the replicates? These experiments are notoriously hard to do and have a strong background, it is therefore important for the community to include all the information in the methods section. In addition, in Figure 1c error bars from independent experiments are missing.

- Figure 3B: The authors suggest that “We identified about 8,000 ZNF207 binding sites across the genome of hESCs, with the 180 strongest binding clustered around the transcription start site (TSS)”. ChIP-Seq is a great technique to interrogate the binding of proteins to DNA. However, it has many caveats and one is that it often can give false positive results, particularly at open chromatin and at highly transcribed regions (see for example here: <http://www.pnas.org/content/110/46/18602.full.pdf>). To overcome these limitations is very hard, but it is important to keep this in mind when stating that the strongest binding occurs at TSSs. The authors should check whether these regions are more commonly chipped compared to other regions (as so-called HOT regions), whether they are particularly strongly expressed and whether expression of these genes is indeed affected upon KD.

- The authors find a strong overlap with Oct4 and Sox2 binding across the genome and present these data as a Venn diagram (Figure 4c). First, this diagram should be scaled like the one presented in Figure 3d. When calling ChIP peaks we always use cutoffs and a real overlap could be bigger. Therefore, many people are using heatmaps to show the simultaneous binding of two proteins. The authors did that too, but only in figure 5a. The authors should use this as well in figure 4. Maybe it would be even good to combine these two figures in some ways. This would also help with Figures 4e and f. The authors show here examples where Oct4, Sox2 and Znf207 co-bind. However, for some of these regions, it is not very clear whether the binding indeed overlaps (see Sall4 for example).

- Figure 5a: The authors show aggregation blots for H3K27ac and below heatmaps. Here it is very curious that the H3K27ac mark shows a peak that overlaps with the p300 mark. This is rather unusual, since p300 is normally binding in open chromatin, here the histones are removed and so the mark is normally found distributed in two peaks besides the p300 mark. Are the authors sure that this is indeed an H3K27ac ChIP?

- Figure 5d: a better annotation of the genes would be useful. For example, are the genes upregulated in the KD specific for a certain cell cycle phase or do they represent senescent cells?

- Figure 6b: It is unclear from overall cell numbers to determine what happened to the actual cells. Would it not be better to normalize these numbers to all cells counted and then present % of cells that show the different markers? In the current form, it could just be lower cell number due to cell cycle effects.

Minor points:

- Figure 1: Where exactly is the TAL effector domain targeted, annotated together with the primers as shown in Figure 1b. In Figure S1a, the targeted locus seems to be directly within the proximal enhancer, however, the ChIP results in figure 1b suggest that it binds 3' of the proximal enhancer.

- Figure 1: The figure used to depict the Pou5F1 gene locus is a bit confusing. Normally people are using for exons bigger and 3' and 5' UTRs smaller boxes. Here all of a sudden the gene has untranslated parts in the middle?

- Figure 1c, 3e, 4d and 5b lack information: what are the red bars and what are the blue bars?

- The references need rework, some do not fit in the context (for example the following: "ZNF207 mRNA is the target of a splicing factor SFRS11 and during somatic reprogramming, ZNF207 changes from isoform B to A and C32". Meant is reference 33, I assume, but that should be checked before submission).

- All examples of ChIP-seq need a scale bar to indicate the size of the chromosome shown.

- Figure 3c should indicate the Distal enhancer of Oct4 that is clearly strongly bound by Oct4.

- Figure S5 needs rework, please compare to Figure 3c, Oct4 changed its TSS in the figure, as did Sall4 (when compared to Figure 4e).

- siRNA sequences should be indicated, preferentially in combination with a depiction of the RNA and where they target (in the text it is said that Table S6 contains this information, but it does not).

- Primer sequences for the different isoforms should be depicted in the RNA and shown how the different isoforms were differentiated from each other by qPCR in Figure 7.

- It should be indicated how the overexpression form is protected from the KD of the siRNAs.

- The methods section needs rewriting. The authors probably did not use 1mg/ml Puromycin (under transfection) and did probably also not use 12mg of lentiviral vector DNA for their virus generation.

- Scale bars in all microscopy images are missing.

Reviewer #3 (Remarks to the Author):

The manuscript "A distinct isoform of ZNF207 controls self-renewal and pluripotency of human embryonic stem cells in conjunction with master transcription factors" identifies new regulators of Oct4 in conventional human ESC and further characterises ZNF207 in gain- and loss-of-function experiments. The authors use an unbiased genome targeting approach based on TALEN mediated binding at the proximal enhancer of Oct4 combined with subsequent Mass spectrometry. They detect previously reported regulators of Oct4 and extensively validate new interactors. The manuscript is well written and the experimental approach conclusive, resulting in comprehensive characterisation of a new primed pluripotency factor. Moreover, the authors provide a rich resource of additional OCT4 interactors. This study is certainly of interest for the pluripotency field, however would substantially benefit from the following points:

1. Can ZNF207 KD be rescued by constitutive OCT4-transgene expression? This addresses the

question whether ZNF207 primarily acts through OCT4 or whether there are other important ZNF207 targets...

2. What about the self-renewing capacity of ZNF207 overexpressing hESC? Can they self-renew for prolonged time in the absence of either bFGF or Activin or both? This is best done by clonogenicity assays after culture in experimental conditions.
3. The authors use conventional human ESC, which have recently been shown to correspond to pregastrula stages of the primate embryo (Nakamura et al., Nature 2016). It would be important to clarify this in the results part where the authors talk about ZNF207 expression in the human preimplantation embryo. Moreover, it would be informative to establish the expression profile of ZNF207 in the primate postimplantation embryo from Nakamura et al, 2016.
4. Recently, naïve pluripotent human ESC have been reported. Two of these protocols (t2iLIF+Go or 5i/LIF) seem to establish some features of an earlier, preimplantation developmental state. It would be interesting to assess a requirement for ZNF207 in the naïve state, e.g. knockdown and subsequent clonogenicity assay in naïve conditions.
5. For the three germ layer differentiation experiments, it is unclear how long and how complete ZNF207 remains depleted in the course of the differentiation protocol. It is essential to assess and report ZNF207 levels in a time course throughout the in vitro differentiation.
6. The IF images in Fig. 6D are rather poor, SOX2 is not visible in any of them, including controls. It would be nice to have larger magnification and confocal images, potentially with insets...
7. Lines 247 – 252 ought to be in the discussion.

We thank the reviewers for their helpful comments. In this revised version of manuscript, we have incorporated additional experimental data and analysis to respond to each of the reviewers' comments. We have highlighted the revisions in yellow in the main text. We believe that this substantially revised manuscript have addressed all the reviewers' concerns.

-Follow the directions of Reviewer 2 to strengthen conclusions that can be drawn about the specific role of ZNF207 as a transcription factor – such as by confirming its role in somatic cells, as well as its role in DNA binding

Response: We have provided additional data based on Reviewer 2's comments.

Importantly, we have tested the direct binding of ZNF207 protein with the potential DNA motif that we predicted from ChIP-seq data by EMSA (electrophoretic mobility shift assay) (Fig. 3d, e). As shown, EMSA with the wild-type probe detected specific ZNF207 protein/DNA complex formation, while mutation of the specific nucleic acids within the motif abolished the binding. This EMSA analysis strongly supports the conclusion that ZNF207 acts as a transcription factor to recognize specific DNA sequences and directly bind to DNA.

Below is a summary of all of the data presented in the manuscript in support of the conclusion that ZNF207 acts as a transcription factor in hESCs:

- 1) Nuclear staining of ZNF207 in hESCs (Fig. 2d);
- 2) Chromatin immunoprecipitation (ChIP) analyses that demonstrate the distribution of its binding in the genome peaking at the promoters of genes, which is typical for transcription factors (Fig. 3b, c; Fig. 4b; Fig. 5a);
- 3) Direct and specific binding of ZNF207 protein to its specific DNA motif by EMSA. Mutation of the specific nucleic acids within the motif abolished the binding (Fig. 3d, e);
- 4) Binding to the regulatory sequences (enhancer) to stimulate gene transcription (Supplementary Fig. 3a)

Together, these data represent the results of the most complete set of experiments possible to prove that ZNF207 acts as a transcription factor in hESCs.

In addition, we also performed chromatin alignment analysis in HeLa cells with control siRNA and ZNF207 KD siRNAs. We validated the alignment of the chromosomes as instructed in the literature (Jiang et al., 2014; Toledo et al., 2014) (Supplementary Fig. 7a, b). Moreover, consistent with the published data, ZNF207 KD caused misalignment of the chromosomes in HeLa cells. This suggests that the KD condition and analysis of chromosome alignment works well in our hands. We observed the function of ZNF207 in chromosome alignment in HeLa cells, but not in hESCs, indicating that a distinct isoform of ZNF207 plays different roles in hESCs.

-Confirm that Znf207 does not simply play a role in regulating cell proliferation.

Response: We have observed the proliferation rate of the cells during the process of reprogramming and find that the effect of ZNF207 on reprogramming efficiency is not just due to its role in regulating cell proliferation.

We have measured the proliferation of fibroblast in the condition of control, ZNF207 KD and ZNF207 OE during a period of 15 days, which is comparable to the time course of somatic cell reprogramming (Supplementary Fig. 2d). We found that KD of ZNF207 only slowed the cell proliferation rate to approximately 75% of that of the control, while OE of ZNF207 had no obvious effect on cell proliferation. However, ZNF207 KD reduced the reprogramming efficiency by approximately 10-fold and ZNF207 OE increased the efficiency by about 3-fold (Fig. 2e). Thus, these data suggest that although ZNF207 contributes to proliferation of fibroblasts to some extent, it functions in the reprogramming mainly through other mechanisms. And, moreover, our data indicates that ZNF207 induces the endogenous *OCT4* expression during reprogramming as one of the modulators of reprogramming mechanisms (Fig. 2f).

-Further clarify the role of Znf207 in regulating pluripotency as directed by Reviewer 3, such as by determining the effects of Znf207 on hESC maintenance in the absence of Activin and bFGF.

Response: In addition to our previous data on conventional (or primed) hESCs, we have further confirmed the role of ZNF207 as a self-renewal and pluripotency regulator in naïve pluripotent stem cells. Moreover, we have found that ZNF207 is able to promote self-renewal of hESCs in conditions with low bFGF.

We analyzed the role of ZNF207 by knockdown in naïve pluripotent human stem cells and found that it is required for maintenance of naïve human pluripotent stem cells, as well. Depletion of ZNF207 in naïve-state hPSCs resulted in disruption of colony forming morphology (Supplementary Fig. 3d), loss of alkaline phosphatase staining (Supplementary Fig. 3e) and reduced expression of pluripotency genes (Supplementary Fig. 3f). These results suggest that ZNF207 is a crucial factor in maintaining self-renewal and pluripotency in both conventional (or primed) hESCs (comparable to post-implantation epiblast) and naïve state hPSCs (comparable to pre-implantation epiblast).

We also observed that *FGF2* (*bFGF*) is one of the direct targets of ZNF207: ZNF207 binds to the promoter and enhancer regions of the *FGF2* gene (Fig. 5f). The expression level of *bFGF* is correlated with the level of ZNF207 (Fig. 5g, h). Furthermore, ZNF207 OE is shown to promote the production of bFGF from the cells and thus greatly prolongs self-renewal of hESCs in low concentrations of bFGF (Fig. 5i).

-Incorporate controls, annotation, improved microcopy, as well as clarify methods used in this manuscript, as specified by all reviewers to provide stronger support for the conclusions drawn.

Response: Based on the reviewers' comments, we have incorporated controls and annotations for all our experiments and improved the resolution of all the images in the figures. The data and conclusions have been greatly improved with the addition of new data and analysis.

We ask that you examine this revised version of the paper, which has addressed all concerns. Again, we thank the editors and reviewers for their consideration.

Reviewer #1 (Remarks to the Author):

Despite extensive studies in understanding self-renewal and pluripotency in mouse embryonic stem cells (ESCs), relatively less is known about how self-renewal and pluripotency is regulated in human ESCs. In this study, Fang et al. performed a locus specific proteomics study to identify novel endogenous regulators of proximal enhancer (PE) of OCT4 in hESCs and identified a number of previously unappreciated factors. Then they focused on one candidate, ZNF207 for detailed mechanistic studies with genomic approaches such as ChIP-seq and RNA-seq analyses. They found that ZNF207 interacts with master TFs (OCT4/SOX2) to maintain hESC self-renewal and pluripotency, and enhances somatic cell reprogramming efficiency through induction of endogenous OCT4 expression. They also found that ZNF207 mainly controls cell cycle regulators. Like other master TFs, they discovered that ZNF207 can function as ectoderm lineage driver through direct binding and regulation of neuronal TFs. Finally, they found that the distinct role of ZNF207 during differentiation occurs via the isoform switching. While manuscript is clearly written and data well presented in supporting for the conclusions drawn, the technique used and some findings are not novel. Many similar locus specific proteomics approaches have been published such as the one in mouse ESCs¹. And ZNF207 isoform switch in pluripotency control has been reported during the reprogramming process². Nonetheless, the study does provide additional insights into our understanding of human ESC pluripotency by integrating ZNF207 into the core pluripotency regulatory network. I have the following critiques that can further enhance the clarity and impact of the current story.

References

1 Rafiee, M. R., Girardot, C., Sigismondo, G. & Krijgsveld, J. Expanding the Circuitry of Pluripotency by Selective Isolation of Chromatin-Associated Proteins. *Molecular cell* 64, 624-635, doi:10.1016/j.molcel.2016.09.019 (2016).

2 Toh, C. X., Chan, J. W., Chong, Z. S., Wang, H. F., Guo, H. C., Satapathy, S., Ma, D., Goh, G. Y., Khattar, E., Yang, L., Tergaonkar, V., Chang, Y. T., Collins, J. J., Daley, G. Q., Wee, K. B., Farran, C. A., Li, H., Lim, Y. P., Bard, F. A. & Loh, Y. H. RNAi Reveals Phase-Specific Global Regulators of Human Somatic Cell Reprogramming. *Cell Rep* 15, 2597-2607, doi:10.1016/j.celrep.2016.05.049 (2016).

Response: We appreciate the positive reflection on the manuscript and take this opportunity to summarize the novelty of our manuscript in order to stress the significance of our findings and then we address each point addressed. This manuscript reports:

- 1) Optimized locus-specific proteomics that provides a genome-wide, unbiased approach of cataloging master regulators at a given chromosomal locus. We like to emphasize that our method is to dissect the endogenous proteomics at a specific chromosomal locus, while the method in reference 1 (Rafiee, M. R. et al., 2016) is to analyze the interacting protein partners with a specific protein. In addition to ChIP and mass spectrometry that have been used in both methods, our method include gene targeting strategies (eg., TALEN/CRISPR) to enable proteomic analysis in the resolution of one specific locus.
- 2) Identification of novel endogenous master regulators of *OCT4* in hESCs.
- 3) Documentation that ZNF207 is a member of the core transcriptional network in hESCs and regulates self-renewal and pluripotency.
- 4) Data that indicates that ZNF207 enhances reprogramming efficiency by induction of endogenous *OCT4* expression.
- 5) Data that shows that ZNF207 promotes cell cycle progression of hESCs.

- 6) Data that indicates that ZNF207 controls commitment to ectoderm differentiation.
- 7) Different isoforms and functions of ZNF207 occur in different cell types. As mentioned by the reviewer, different isoforms of ZNF207 have been identified (Toh, C. X. et. al., 2016). However, this work, for the first time, has found that different isoforms of ZNF207 have played distinct roles in various cell types. This is really one of the most interesting findings in our study, which suggests that a protein may change its functions in the development by switching the isoforms.
- 8) Isoform C of ZNF207 is the functional isoform in hESCs.

Main critiques:

1. Since the proteomics screen identified ~150 proteins, it would be nice to have a broader functional test of candidates by loss of function studies besides ZNF207 to show the robustness of the approach in identification of novel factors with functional significance in controlling human pluripotency, then narrow down to ZNF207 for a detailed mechanistic dissection. The authors could utilize previously published RNAi screens for the maintenance and exit of pluripotency (From the Huck hui Ng Lab publications) to evaluate how many of their candidates are present in the previous screens. ZNF207 may be present already in those previous screens. Then pick some of them that are not present in previous screens and test them here by shRNA/siRNA experiments.

Response: We have compared our list (152 proteins) with the previously published RNAi screen list (565 genes) (Chia. et al., 2010). We identified just 9 targets that are present in common and ZNF207 is not one of them (see Figure 1 below).

Figure 1. Venn diagram of the genes identified by our locus-specific proteomics and genes identified by whole genome RNAi screening (Chia. et al., 2010).

Low overlap between these two lists is due to the completely different screening goals, readouts and strategies that were used. Our goal of screening is to identify endogenous proteins that directly regulate *OCT4* expression through its proximal enhancer, while the goal of the other group is to identify genes whose downregulation would cause change in *OCT4* expression. The strategy we used is genome-targeting, chromatin immunoprecipitation and proteomics, while the other group is whole genome RNAi screen. The readouts of our study are the endogenous

proteins that are associated with the *OCT4* enhancer, while the readouts of the other study are the GFP signals driven by the *OCT4* upstream regulatory region. Essentially, the targets in our list are proteins that are associated with *OCT4* enhancer. Some of them could be direct regulators of *OCT4*, but some of them could also associate with this specific locus through protein-protein interactions. The ultimate goal of our method is to develop a locus-specific proteomics to identify potential endogenous regulators for a specific gene. However, the targets in the other list could be *OCT4* regulators, but may not function through the *OCT4* proximal enhancer. Or their targets may disrupt self-renewal and pluripotency of hESCs through other mechanisms or pathways, and then induce loss of cell identity, which indirectly affect *OCT4* expression.

Interestingly, ZNF207, which has been identified and verified as essential hESCs and *OCT4* regulators by our screening and functional analyses, was not identified by the previous RNAi screening. This suggests that our method of locus-specific proteomics could be used as a great addition or complement to RNAi screen to identify novel hESCs regulators. Moreover, it can also be theoretically applied to any specific locus to identify regulators for any particular gene or locus of interest.

In addition to ZNF207, we selected another five genes for siRNA analysis. These five genes are not listed in the other RNAi screen and KD of each of these results in a change of *OCT4* expression, suggesting that they are potential *OCT4* regulators (Supplementary Fig. 2b).

2. The loss of function of ZNF207 in compromising reprogramming could be due to the defect on somatic cell proliferation during the reprogramming. Does ZNF207 KD affect somatic cell growth? Also, based on the Loh study², the isoform switch in pluripotency has been detailed during the reprogramming process, does the KD in this study abrogate all the isoforms or specific isoform of ZNF207 during the reprogramming? In other words, can isoform specific shRNA be tested in reprogramming?

Response: We have assessed the proliferation of fibroblasts in the conditions of control, ZNF207 KD and ZNF207 OE over a period of 15 days, which is comparable to the time course of somatic cell reprogramming (Supplementary Fig. 2d). We found that KD of ZNF207 only slowed the cell proliferation rate to about 75% compared to the control and OE of ZNF207 had no obvious effect on cell proliferation. However, ZNF207 KD reduced the reprogramming efficiency to about 10-fold and ZNF207 OE increased the efficiency to about 3-fold (Fig. 2e). These data suggest that although ZNF207 controls the proliferation of fibroblasts at a basal level, it affects the reprogramming mainly through other mechanisms. And our data indicate that ZNF207 induces the endogenous *OCT4* expression during reprogramming is one of the mechanisms (Fig. 2f).

For the KD, we abrogated all the isoforms of ZNF207 during the reprogramming process. As presented in Fig. 7c, isoform C, which is the functional isoform in hESCs, is the longest isoform. Any siRNAs designed to target the other two isoforms will abrogate isoform C. So, it is not possible to test isoform specific KD effect in the reprogramming process. We have made the change of the text on the page 5 of the main text, highlighted in yellow.

3. Fig. 4a/b shows the composition of OCT4/SOX2 motif, but the colP is only validated for OCT4. How about the SOX2-ZNF207 interaction?

Response: In this revised manuscript, we have tested the protein interaction between ZNF207 with SOX2 by co-immunoprecipitation, as well. We found that ZNF207 is able to pull-down

SOX2 in hESCs and *vice versa* (Fig. 4c), suggesting that ZNF207 may form a protein complex with both OCT4 and SOX2 in hESCs.

We have made the change in the text on the page 7 of the main text, highlighted in yellow.

Minor points:

1. Line 102: DNaseI should be DNaseI

Response: Thank you for pointing out the mistake. We have corrected it on page 4 of the main text, highlighted in yellow.

2. Line 280: ZNF07 should be ZNF207

Response: Thank you for pointing out the mistake. We have corrected it on page 10 of the main text, highlighted in yellow.

3. Line 288: Figure7c should be Figure 7c with a space.

Response: Thank you for pointing out the mistake. We have corrected it on page 10 of the main text. Based on the instructions of Nature Communications, we have cited all our Figures as Fig. and supplementary figures as Supplementary Fig.

4. Line 293: Isoform should be isoform.

Response: Thank you for pointing out the mistake. We have corrected it on page 10 of the main text.

5. Line 294-297: Figure S6a/b should be Figure S7a/b.

Response: Thank you for pointing out the mistake. We have corrected it on page 10 of the main text.

6. I don't understand two places why ref#32 was cited: 1) Line 290; 2) Line 342. The ref#32 has nothing to do what were mentioned there.

Response: Thank you for pointing out the mistake. We disrupted the order of the references while manually edited it in the previous version of manuscript. It should be the reference below:

Toh CX, et al. RNAi Reveals Phase-Specific Global Regulators of Human Somatic Cell Reprogramming. *Cell Rep.* 2016;15(12):2597-607.

We have made sure that all the references are correctly ordered and referred in the text in this revised version of manuscript.

7. Fig. 1c, 3e, 4d: it would be nicer if color coded text is used to match the colored bars. For example, in Figure 4d: "Gene number" in blue and "p-value" in orange to match the same color bars.

Response: Thank you for the suggestion. We have used the color-coded text to match the colored bars for all our GO analysis in Fig. 1c, 3g, 4e and 5d.

8. Fig. 4b: the western blots are too grainy. Not sure if Photoshop background reduction or what.

Response: We repeated the western blot and the new data, as is, is presented in Fig. 4c.

Reviewer #2 (Remarks to the Author):

In the paper “A distinct isoform of ZNF207 controls self-renewal and pluripotency of human embryonic stem cells in conjunction with master transcription factors” the authors set out to understand the regulation of OCT4 in human pluripotent stem cells. For this, they first established a proteomic approach based on the proximal enhancer of OCT4 and isolated proteins that could bind to this enhancer. Through mass spec, the authors identified known and unknown factors that might be involved in the regulation of OCT4. The authors focused on ZNF207 for the remainder of the paper and studied the function of this protein in primed pluripotency. One of the most exciting findings of the paper is that ZNF207 has different isoforms that show vastly different functions. While ZNF207 has been extensively studied in the context of kinetochores and loss of it interferes with spindle assembly in somatic cells, in human embryonic stem cells a different isoform seems to be expressed that does not interfere with spindle assembly but instead affects pluripotency program as a transcription factor.

The functions of Znf207/BugZ has been in studied in mitosis: ZNF207 is important for the spindle assembly in somatic cells (Toledo et al. Dev Cell, 2014). In addition, ZNF207 is involved in splicing in pre-mRNA (Wan et al. JCB, 2015).

ZNF207 had been previously shown to affect reprogramming efficiency (see Toh et al, 2016, Cell Reports) and it was also shown that ZNF207 has different isoforms that affect reprogramming in different forms. However, the underlying mechanism was still unclear. The authors here set out to analyze the underlying function of ZNF207 in pluripotent stem cells and find that it might be acting as a transcription factor. While the conclusions are very exciting and of interest for the community, they are not supported by the data presented and in its current form, the manuscript would not warrant publication in Nature Communications. In particular, many controls are missing. However, the manuscript can be strengthened by addressing points laid out below.

General Comment:

Overall, I am very disappointed with the quality of parts of the manuscript. For example, the microscopy images shown in figures 2d, 5e, 6a and d are all very low quality and reveal very little detail of the cells (for example, the authors claim that TRA1-60 is lost from the cell membrane and now found in the cells. That is not very clear from the data presented). If possible, the authors should try to find better examples or maybe repeat some of the experiments. In the current form, the images do not show very much.

Response: We have remedied the overall quality of the images in the previous version of the manuscript. The images have lost much of the resolution when we reduced the size of the PDF files for submission. To solve this, we re-created the images with higher magnification and resolution, and increased the resolutions for all the images when we compressed the PDF files. The overall quality of the images has been much improved in this revised manuscript, please refer to the new images for Fig. 2d, Supplemental Fig. 5c (previous Fig. 5e) and 6a.

Major Points:

- The author's main conclusion is that ZNF207 acts as a transcription factor. These data are supported by ChIP-seq experiments and knock down experiments. Both techniques are not able to conclusively show that it is actually a transcription factor in addition to its roles in mitosis. For example, does ZNF207 bind DNA directly? The data shown suggest that it localizes to DNA, but it does not show an affinity for DNA. In these cases, I would

expect either a mutational analysis of ZNF207, identification of the DNA binding domain or a direct readout of DNA-binding affinity through gel shift assays etc. The authors present a potential motif for DNA binding, but that could also be anything, bound by any other TF in the vicinity of OCT4. A mutational analysis with concomitant loss of the binding would be necessary to further corroborate these findings.

Response: We have tested the direct binding of ZNF207 protein with the potential DNA motif that we predicted from ChIP-seq data by EMSA (electrophoretic mobility shift assay) (Fig. 3d, e). As shown, EMSA with the wild-type probe detected specific ZNF207 protein/DNA complex, while mutation of the specific nucleic acids within the motif abolished the binding. This EMSA analysis strongly support our conclusion that ZNF207 acts as a transcription factor to recognize specific DNA sequences and directly bind to DNA. We have made the change of the text on the page 6-7 of the main text, highlighted in yellow.

Here, we summarize the data presented in the manuscript in support of the conclusion that ZNF207 acts as a transcription factor in hESCs which includes:

- 1) Nuclear staining of ZNF207 in hESCs (Fig. 2d);
- 2) Chromatin immunoprecipitation analyses show the distribution of its binding in the genome peaking at the promoters of genes, which is typical for transcription factors (Fig. 3b, c; Fig. 4b; Fig. 5a);
- 3) Direct and specific binding of ZNF207 protein to its DNA motif by EMSA. Mutation of the specific nucleic acids within the motif abolished the binding (Fig. 3d, e);
- 4) Binding to the regulatory sequences (enhancer) to stimulate transcription of the gene (Supplementary Fig. 3a)

Together, these data represent the results of the most complete set of experiments possible to prove that ZNF207 acts as a transcription factor in hESCs.

- Figure 7: The most interesting finding of the manuscript is that the function of Znf207 in pluripotent stem cells seems to be different from its function in somatic cells. For this, the authors perform immunofluorescence of Tubulin, CREST in combination with DAPI in WT and in Znf207KD cells. In pluripotent cells, the KD did not affect the alignment of the chromosomes and the authors conclude that the function of Znf207 in pluripotent stem cells is different. However, these conclusions have to be supported by data showing that under the conditions that the authors are working, Znf207KD in somatic cells does indeed affect chromosome alignment. The authors used siRNAs that were ordered from a company. To publish such a negative result (no misaligned chromosomes) it is necessary to show that the underlying experimental conditions can support their conclusions.

Response: We used the Silencer Select siRNAs from Ambion for our knockdown experiments. Three siRNAs that target ZNF207 were transfected together for high efficiency of knockdown. KD of ZNF207 were confirmed by Real-time RT-PCR and western blot. The mRNA and protein of ZNF207 has been reduced to about 5-10% (Supplementary Fig. 2c). We have made the change of the text on the page 5 of the main text, highlighted in yellow, as well.

To test validity of the chromatin alignment analysis, we performed ZNF207 KD in HeLa cells and checked the alignment of the chromosomes as instructed in the literature (Jiang et al., 2014; Toledo et al., 2014). Consistent with the published data, ZNF207 KD caused misalignment of the chromosomes in HeLa cells (Supplementary Fig. 7a, b). We have made the change of the text on the page 10 of the main text, highlighted in yellow.

These results suggest that the KD condition and analysis of chromosome alignment works well in our hands. We did observe the function of ZNF207 in chromosome alignment in HeLa cells, but not in hESCs, indicating that a distinct isoform of ZNF207 plays different roles in hESCs.

- Furthermore, a second role described for Znf207 was shown to be its role in splicing and downstream from this R-loop formation. Did the authors consider this possibility and exclude it in this manuscript?

Response: We thank the reviewer for the suggestion and note that this was not excluded on purpose. We have tested whether ZNF207 plays a role in splicing and formation of R-loop in hESCs as reported in somatic cells (Wan et al., 2015). We knocked down ZNF207 in both HeLa and hESCs and stained for R-loop by the monoclonal antibody S9.6 (ENH001; Kerafast, Inc). We found that depletion of ZNF207 stimulates R-loop formation in HeLa cells (Figure 2a, below), however, no obvious R-loop is observed in hESCs (Figure 2b, below). We also analyzed RNA-seq data and we found that depletion of ZNF207 did not result in significant change of splicing events when compared to the control siRNA KD. Below, we show three representative genes, including *WAC*, the gene that was studied in the reference (Wan et al., 2015). KD of ZNF207 did not cause splicing defects (either intron retention or exon skipping) for these three genes in hESCs (Figure 2c, below). These results suggest that depletion of ZNF207 does not change the splicing and R-loop formation in hESCs. This may be due to the distinct function of individual isoform of ZNF207 present in different cell types.

Figure 2. Role of ZNF207 in R-loop formation and splicing in hESCs. (a) Immunofluorescence staining of R-loop and DAPI in ZNF207 KD HeLa cells. Scale bar, 5 μ m. (b) Immunofluorescence staining of R-loop and DAPI in ZNF207 KD hESCs. White broken circles outline the nuclei with no or low levels of R-loops. Scale bar, 5 μ m. (c) RNA-seq data from control siRNA or ZNF207 siRNA transfected hESCs. Three representative genes, *WAC*, *SALL4* and *NANOG*, are shown. Scale bar of the size of the genome is shown.

- Figure 1: It is unclear what kind of controls were used in the initial proteomic analysis? Did the authors include a negative region such as a TALEN designed towards an inactive region? How many independent experiments were performed and how much overlap existed between the replicates? These experiments are notoriously hard to do and have a strong background, it is therefore important for the community to include all the information in the methods section.

In addition, in Figure 1c error bars from independent experiments are missing.

Response: We have used a TALE construct that targets GFP (green fluorescence protein) as a negative control to eliminate unspecific binding. The experiments were independently performed twice and the overlap was approximately 85%. We provide the detailed information in the method on page 16-18 of the main text.

As in other screening experiments such as whole genome RNAi, our locus specific proteomic screening may have background noises due to the technical complexity of the experiments. However, the purpose of these experiments is to provide a novel resource to identify targets of interest. Following validation, functional assays and integrative analyses with other database (eg., proteomic, transcriptomic and knockout phenotype) will further narrow down the list to sort out the final targets of interest. Essentially, we provide a method and resource to identify critical regulators that have not been uncovered by other existing methods. In this study, after proteomics, we performed ChIP and knockdown analyses on several candidates in the list to validate their binding on the locus and their potential regulatory roles for *OCT4* gene (Supplemental Fig. 2a, b). Then we chose to investigate ZNF207, which gave us the strongest phenotype after KD, with more detailed functional analysis. Together, these results demonstrated the robustness of the method and ability to identify novel putative regulators of OCT4 that may function in maintaining self-renewal and pluripotency of hESCs.

For the gene ontology analysis in Fig. 1c in the manuscript, we used the list comprised of 152 proteins for the analysis. These 152 proteins are the proteins that overlapped between two independent proteomic studies. In most cases, GO analysis is done using one list of proteins without error bars. Instead, p-values are provided to show the significance of a particular GO term that is associated with the list of proteins.

- Figure 3B: The authors suggest that “We identified about 8,000 ZNF207 binding sites across the genome of hESCs, with the 180 strongest binding clustered around the transcription start site (TSS)”. ChIP-Seq is a great technique to interrogate the binding of proteins to DNA. However, it has many caveats and one is that it often can give false positive results, particularly at open chromatin and at highly transcribed regions (see for example here: <http://www.pnas.org/content/110/46/18602.full.pdf>). To overcome these limitations is very hard, but it is important to keep this in mind when stating that the strongest binding occurs at TSSs. The authors should check whether these regions are more commonly chipped compared to other regions (as so-called HOT regions), whether they are particularly strongly expressed and whether expression of these genes is indeed affected upon KD.

Response: We thank the reviewer for the comments on ChIP technique and the suggestions. The intention was to investigate the distribution of ZNF207-binding peaks relative to TSS to determine whether its binding forms peaks at the promoters of genes, which is one of the major characteristics for transcription factors. This is one piece of evidence that supports the conclusion that ZNF207 acts as a transcription factor in hESCs.

As to the HOT regions, we realize that there could be false positive signals in any of the ChIP-seq studies. To filter out the noise or false positive signals, genomic fragments from input and an IgG control ChIP were also sequenced for potential biases or false positives in the distribution of genomic DNA fragments. We then combined RNA-seq data from KD with our ChIP-seq data. As shown in Fig. 3f, we focused on differentially expressed genes that are also bound by ZNF207 as true targets for ZNF207 for the downstream analysis.

- The authors find a strong overlap with Oct4 and Sox2 binding across the genome and present these data as a Venn diagram (Figure 4c). First, this diagram should be scaled like the one presented in Figure 3d. When calling ChIP peaks we always use cutoffs and a real overlap could be bigger. Therefore, many people are using heatmaps to show the simultaneous binding of two proteins. The authors did that too, but only in figure 5a. The authors should use this as well in figure 4. Maybe it would be even good to combine these two figures in some ways. This would also help with Figures 4e and f. The authors show here examples where Oct4, Sox2 and Znf207 co-bind. However, for some of these regions, it is not very clear whether the binding indeed overlaps (see Sall4 for example).

Response: 1) We have re-constructed the Venn diagram based on the scale in Fig.4d.
2) We have provided the heatmap in Fig. 4b in order to show the co-binding of OCT4, SOX2 with ZNF207;
3) We have highlighted the co-bound regions by these three proteins by light pink boxes to make the images clearer. In addition, we replaced the original SALL4 binding images with the new zoomed-out images, which provides a better illustration of co-bound regions by OCT4, SOX2 and ZNF207.
4) We would like to keep Fig. 4 and Fig.5 separate, if possible, because Fig. 4 shows co-binding of ZNF207 with OCT4, SOX2 and ZNF207 on pluripotency related genes, while Fig. 5 shows co-binding of ZNF207 with P300 and H3K27ac on cell cycle genes. These are two characteristics of ZNF207 binding in the genome of hESCs, which suggest two distinct functions of ZNF207 in hESCs.

- Figure 5a: The authors show aggregation blots for H3K27ac and below heatmaps. Here it is very curious that the H3K27ac mark shows a peak that overlaps with the p300 mark. This is rather unusual, since p300 is normally binding in open chromatin, here the histones are removed and so the mark is normally found distributed in two peaks besides the p300 mark. Are the authors sure that this is indeed an H3K27ac ChIP?

Response: We thank the reviewer for highlighting this. H3K27ac (GSM733718) and p300 (GSM1003513) ChIP-seq data is obtained from published ENCODE data. Our previous representation of these ChIP-seq data was positioned relative to the peaks of ZNF207 ChIP-seq data. Here, we established the position profile of these ChIP-seq data relative to the TSS in Fig. 5a. It is now easier to distinguish that H3K27ac has two peaks besides the P300 peaks (Fig. 5a).

- Figure 5d: a better annotation of the genes would be useful. For example, are the genes upregulated in the KD specific for a certain cell cycle phase or do they represent senescent cells?

Response: We have performed gene ontology analysis for the cell cycle genes that were differentially expressed in control and ZNF207 KD cells. We found that genes involved in senescence, cell cycle checkpoint and cell cycle phase transition are upregulated, and genes involved in regulation of cyclin-dependent protein kinase (CDK) activity, telomerase activity and

DNA damage response are downregulated in ZNF207 KD cells (Fig. 5d). This is consistent with the findings that pluripotent cells exhibit high CDD and telomerase activity, lack checkpoint regulation and do not appear to undergo senescence. When differentiation occurs or pluripotency is lost, cells become cell-cycle regulated and their DNA repair ability is decreased. We have made the change of the text on the page 8 of the main text, highlighted in yellow.

- Figure 6b: It is unclear from overall cell numbers to determine what happened to the actual cells. Would it not be better to normalize these numbers to all cells counted and then present % of cells that show the different markers? In the current form, it could just be lower cell number due to cell cycle effects.

Response: We thank the reviewer for the suggestion. We include both overall cell number and the percentage of the cells (normalized number) that are stained positive for markers in Fig. 6b. We seeded the same number of control and KD cells for specific lineage differentiation. The overall cell number shows the proliferation and survival rate of the cells in the differentiation condition (we did observe extensive cell death of ZNF207 KD cells in the ectoderm differentiation media), while the percentage of cells that are stained positive for markers shows the differentiation efficiency of the survived cells.

Minor points:

- Figure 1: Where exactly is the TAL effector domain targeted, annotated together with the primers as shown in Figure 1b. In Figure S1a, the targeted locus seems to be directly within the proximal enhancer, however, the ChIP results in figure 1b suggest that it binds 3' of the proximal enhancer.

Response: The exact TALEN targeted sequences are listed in Supplementary Table 1 and are within the proximal enhancer. Unfortunately, the label on the ChIP primers was not accurate in the previous version of manuscript, we have corrected this in Fig. 1b to show the accurate position of the ChIP primers.

- Figure 1: The figure used to depict the Pou5F1 gene locus is a bit confusing. Normally people are using for exons bigger and 3' and 5' UTRs smaller boxes. Here all of a sudden the gene has untranslated parts in the middle?

Response: We thank the reviewer for the suggestion. As this study is focused on the regulatory elements of *OCT4* gene, rather than the gene body, we have simplified the presentation of the gene body to better depict the regulatory elements. Please see Fig. 1a, b for the change.

- Figure 1c, 3e, 4d and 5b lack information: what are the red bars and what are the blue bars?

Response: We have taken the advice from Reviewer #1, as well, and used the color-coded text to match the colored bars in Fig. 1c, 3g, 4e and 5d for better presentation. Also, we put the information of colored-bars in the figure legend on the page 13-15 of the main text.

- The references need rework, some do not fit in the context (for example the following: "ZNF207 mRNA is the target of a splicing factor SFRS11 and during somatic reprogramming, ZNF207 changes from isoform B to A and C32". Meant is reference 33, I assume, but that should be checked before submission).

Response: We thank the reviewer for pointing out the mistake. We disrupted the order of the references in editing. We have made sure that all the references are correctly ordered and referred in the text in this revised version of manuscript.

- All examples of ChIP-seq need a scale bar to indicate the size of the chromosome shown.

Response: We have inserted the scale bars to indicate the size of chromosome for each of the genome browser representation of ChIP-seq data. Please see Fig. 3c, 4f, 4g, 5c, 5f and Supplementary Fig. 5a, 6b for the scale bars. The information of the scale bars is also included in the figure legends on the page 13-15 of the main text and page 3 of the supplementary text.

- Figure 3c should indicate the Distal enhancer of Oct4 that is clearly strongly bound by Oct4.

Response: We have indicated the proximal enhancer (PE) and distal enhancer (DE) in the Fig. 3c and the figure legend on the page 14 of the main text.

- Figure S5 needs rework, please compare to Figure 3c, Oct4 changed its TSS in the figure, as did Sall4 (when compared to Figure 4e).

Response: We have modified Supplementary Fig. 5 to make sure that the position of TSS for each gene is correct.

- siRNA sequences should be indicated, preferentially in combination with a depiction of the RNA and where they target (in the text it is said that Table S6 contains this information, but it does not).

Response: The sequences and the locations of siRNAs that we used in Supplementary Fig 2b are listed as Supplementary Table 6. For ZNF207 KD, three siRNAs that target ZNF207 were transfected together for high efficient knockdown effect. The siRNA sequences, interrogated sequence, targeted exons and locations are provided in Supplemental Table 7.

- Primer sequences for the different isoforms should be depicted in the RNA and shown how the different isoforms were differentiated from each other by qPCR in Figure 7.

Response: The qPCR primers for detecting individual isoform is provided in Supplementary Table 8.

- It should be indicated how the overexpression form is protected from the KD of the siRNAs.

Response: The ZNF207 overexpression construct is codon-optimized to be protected from the KD. The sequence information is provided as Supplementary Table 9.

- The methods section needs rewriting. The authors probably did not use 1mg/ml Puromycin (under transfection) and did probably also not use 12mg of lentiviral vector DNA for their virus generation.

Response: We thank the reviewer for pointing out these errors. We have rechecked the details of the methods to make sure they are accurate. The puromycin was used at 2ug/ml for the selection and 12 µg lentiviral vector was used for virus generation. Please refer to the page of 16-21 of the main text for the revision.

- Scale bars in all microscopy images are missing.

Response: Scale bars are provided for each of the microscopy images and information is also provided in the figure legend.

Reviewer #3 (Remarks to the Author):

The manuscript “A distinct isoform of ZNF207 controls self-renewal and pluripotency of

human embryonic stem cells in conjunction with master transcription factors” identifies new regulators of Oct4 in conventional human ESC and further characterises ZNF207 in gain- and loss-of-function experiments. The authors use an unbiased genome targeting approach based on TALEN mediated binding at the proximal enhancer of Oct4 combined with subsequent Mass spectrometry. They detect previously reported regulators of Oct4 and extensively validate new interactors. The manuscript is well written and the experimental approach conclusive, resulting in comprehensive characterisation of a new primed pluripotency factor. Moreover, the authors provide a rich resource of additional OCT4 interactors. This study is certainly of interest for the pluripotency field, however would substantially benefit from the following points:

1. Can ZNF207 KD be rescued by constitutive OCT4-transgene expression? This addresses the question whether ZNF207 primarily acts through OCT4 or whether there are other important ZNF207 targets...

Response: As suggested by the reviewer, we tried to rescue ZNF207 KD cells by constitutively expressing OCT4 protein to the endogenous level and we found that OE of OCT4 is able to restore the ZNF207 KD effect. Depletion of ZNF207 resulted in the loss of alkaline phosphatase staining and colony-formed phenotype, while OE of OCT4 could rescue the effect of ZNF207 KD, cells become alkaline phosphatase positive and form colonies (Supplementary Fig. 3b). Gene expression analysis further demonstrated that OCT4 OE rescued the differentiation by re-activation the expression of pluripotency genes, such as *SALL4*, *SOX2* and *NANOG* (Supplementary Fig. 3c). These data strongly suggest that OCT4 is one of the major functional targets of ZNF207 in hESCs. We also include the description of the data on page 6 of the main text, highlighted in yellow.

2. What about the self-renewing capacity of ZNF207 overexpressing hESC? Can they self-renew for prolonged time in the absence of either bFGF or Activin or both? This is best done by clonogenicity assays after culture in experimental conditions.

Response: Based on the analysis of ZNF207 ChIP-seq data, we found that FGF2 (bFGF) is one of the direct targets of ZNF207: ZNF207 binds to the promoter and enhancer regions of FGF2 gene (Fig. 5f); The expression level of bFGF correlates with the level of ZNF207 (Fig. 5g, h). We then sought to test whether ZNF207 OE could prolong the self-renewal ability of hESCs in the absence or low concentration of bFGF by clonogenicity assays. We found that control hESCs fail to survive in 20ng/ml bFGF culture media, however, ZNF207 OE hESCs can proliferate as single cells and form AP-staining positive colonies (Fig. 5i). These results suggest that ZNF207 OE is able to promote the production of bFGF from the cells and thus prolong the self-renewal of hESCs in the low concentration of bFGF. We also include the description of the data on page 8-9 of the main text, highlighted in yellow.

However, we did not see any rescue effect of ZNF207 OE in the absence of Activin.

3. The authors use conventional human ESC, which have recently been shown to correspond to pregastrula stages of the primate embryo (Nakamura et al., Nature 2016). It would be important to clarify this in the results part where the authors talk about ZNF207 expression in the human preimplantation embryo. Moreover, it would be informative to establish the expression profile of ZNF207 in the primate postimplantation embryo from Nakamura et al, 2016.

Response: We have clarified this on the page 6 of the main text. We examined the expression profile of ZNF207 in the primate post-implantation embryo from Nakamura et al., 2016, however, there is no expression data of ZNF207 in their single cell profiling data (GSE74767). We did find Zfp207 expression data in their mouse profiling data, but not primate profiling data. Since the expression of ZNF207 is high across all the stages of preimplantation development (Supplementary Fig. 3g, h) and in hESCs, we suggest that its expression will maintain high post-implantation. Its function in the early embryo development would be very interesting to explore in the future by mouse and primate study.

4. Recently, naïve pluripotent human ESC have been reported. Two of these protocols (t2iLIF+Go or 5i/LIF) seem to establish some features of an earlier, preimplantation developmental state. It would be interesting to assess a requirement for ZNF207 in the naïve state, e.g. knockdown and subsequent clonogenicity assay in naïve conditions.

Response: We analyzed the function of ZNF207 by KD in naïve pluripotent human stem cells and found that it is required for maintenance of naïve human pluripotent stem cells, as well. The naïve state hPSCs were obtained and cultured based on the protocol of Gafni et al., 2013. Depletion of ZNF207 in naïve state hPSCs resulted in disruption of colony forming morphology (Supplementary Fig. 3d), loss of alkaline phosphatase staining (Supplementary Fig. 3e) and reduced expression of pluripotency genes (Supplementary Fig. 3f). These results suggest that ZNF207 is a crucial factor to maintain self-renewal and pluripotency in both conventional hESCs (comparable to post-implantation epiblast) and naïve state hPSCs (comparable to pre-implantation epiblast). The description of the data is included on page 6 of the main text, highlighted in yellow.

5. For the three germ layer differentiation experiments, it is unclear how long and how complete ZNF207 remains depleted in the course of the differentiation protocol. It is essential to assess and report ZNF207 levels in a time course throughout the in vitro differentiation.

Response: To make sure that the level of ZNF207 remained reduced during differentiation, we transfected siRNAs into the cells on day 0 and day 3 of the differentiation process. The level of ZNF207 over the course of differentiation was reduced to approximately 5% of its level in control cells (Supplementary Fig. 6a). The description of the data is included on page 9 of the main text, highlighted in yellow.

6. The IF images in Fig. 6D are rather poor, SOX2 is not visible in any of them, including controls. It would be nice to have larger magnification and confocal images, potentially with insets...

Response: We have improved the resolution of the images and included the scale bars for all the images in the manuscript. In Fig. 6d, we have replaced SOX2 images with confocal images with larger magnification. Scale bars represents 25um are also provided.

7. Lines 247 – 252 ought to be in the discussion.

Response: We have moved the text to the discussion. Please refer to the page 11 of the main text for the revision.

References for the response letter

Chia, N.-Y. Y. *et al.* A genome-wide RNAi screen reveals determinants of human embryonic stem cell identity. *Nature* **468**, 316-320 (2010).

Jiang, H. *et al.* A microtubule-associated zinc finger protein, BuGZ, regulates mitotic chromosome alignment by ensuring Bub3 stability and kinetochore targeting. *Dev Cell* **28**, 268-281, (2014).

Toledo, C. M. *et al.* BuGZ is required for Bub3 stability, Bub1 kinetochore function, and chromosome alignment. *Dev Cell* **28**, 282-294 (2014).

Wan, Y. *et al.* Splicing function of mitotic regulators links R-loop-mediated DNA damage to tumor cell killing. *J Cell Biol.* 209(2):235-46 (2015).

Nakamura, T. *et al.* A developmental coordinate of pluripotency among mice, monkeys and humans. *Nature* **537**, 57-62 (2016).

Gafni, O. *et al.* Derivation of novel human ground state naive pluripotent stem cells. *Nature* **504**, 282-286 (2013).

Reviewers' comments:

Reviewer #1 (Remarks to the Author):

Main points:

The authors are largely responsive to my previous critiques, and the revised manuscript has been improved. However, there are still several issues that need to be addressed.

Main Critiques:

1. Use the OCT4 upregulation as the mechanism to explain the positive role of ZNF207 in promoting reprogramming has a potential pitfall as OCT4 level has to be tightly regulated for pluripotency. Too much OCT4 will be problematic also in pluripotency and reprogramming. Based on the studies from the Silva group and others, many partially reprogrammed cells, i.e., pre-iPSCs have overexpressed OCT4, and a lower than normal WT level of OCT4 actually is beneficial for pluripotency. The authors have to do more analysis and strengthen their claim on the potential mechanistic action of ZNF207 in promoting hiPSC reprogramming.
2. Fig. S2b: HDAC2 and SALL2 siRNA treatment resulted in over 2-fold increase of OCT4 levels. Since this will cause differentiation of mouse ESCs, what happened to hESCs here should be pointed out.
3. Fig. S3: 1) due to the various cocktails for human naïve ESCs, the authors may want to point out in the main text which cocktail they used for the study shown in Fig. S3; 2) The characterization of KD phenotype is quite superficial and limited with only two marker genes NANOG and SALL4; 3) Fig. S3c should be supplemented with the OCT4 expression level to show how much OCT4 OE is as tight control of OCT4 level is critical for pluripotency.
4. Fig. S6c: I failed to understand or appreciate the authors' statement "forced expression of ZNF207 increased the expression of OTX2" based on the bar chart shown. I did not see increased OTX2 expression at all compared with the Control.
5. Fig. S6d: the OTX2 expression level should also be shown to show its overexpression relative to Control.

Minor points:

1. Line 166: KD of ZNF207 KD should be corrected as "KD of ZNF207" or simply "ZNF207 KD".
2. Line 176: why suddenly hPGCs? Must be the typo for "hPSCs".
3. Lin 188: After briefly described the KD effects on naïve hESCs above, the authors may want to point out the RNA-seq and CHIP-seq afterwards were also performed in conventional primed hESCs.
4. Fig. 4 legends was a mess. (c) was presented twice, and c/d legends do not match the data shown. It is such a careless mistake during the revision stage.
5. Fig. 4e: p-value should be explicitly labeled in the figure also.

Reviewer #2 (Remarks to the Author):

Overall, the paper has much improved and specifically the newly added data showing the direct binding of Znf207 to its predicted binding motif does strengthen the point the authors are making and do indeed support the idea that the protein could in fact binding DNA directly and could act as a transcription factor. Therefore, the authors have addressed most of my concerns. However, the authors identified three different isoforms and showed that the Isoform C is the main isoform expressed in human ESC and that this isoform is also the only one that is able to rescue the loss of Znf207. Which isoform did the authors test in their EMSA? Did they test all three isoforms and only

the C isoform worked or did the only test one, unnamed isoform? This should also be indicated for Figure 2e and 2f, 5h and 5i (did the authors test the other isoforms in these assays?).

Reviewer #3 (Remarks to the Author):

The revised version of the manuscript has substantially improved and addresses the key concerns of the reviewers. The authors carried out a substantial amount of additional experiments to corroborate their conclusions. I recommend to accept this version for publication in Nature Communications.

We thank the reviewers for their comments and positive recommendation regarding publication in *Nature Communications*. This work is a unique and important piece of work in the field of stem cell biology and contributes to better understanding of the endogenous molecular mechanisms underlying self-renewal and pluripotency. In this revised version of manuscript, we have incorporated additional data and analysis to respond to the final comments of the reviewers, especially those of Reviewer 1 regarding OCT4 expression levels in the process of reprogramming. We have highlighted the revisions in yellow in the main and supplementary text. In this revision, we have addressed all the reviewers' concerns. Below is the point-by-point response to the list of the reviewers' comments.

Reviewer #1 (Remarks to the Author):

Main points:

The authors are largely responsive to my previous critiques, and the revised manuscript has been improved. However, there are still several issues that need to be addressed.

Main critiques:

1. Use (of) the OCT4 upregulation as the mechanism to explain the positive role of ZNF207 in promoting reprogramming has a potential pitfall as OCT4 level has to be tightly regulated for pluripotency. Too much OCT4 will be problematic also in pluripotency and reprogramming. Based on the studies from the Silva group and others, many partially reprogrammed cells, i.e., pre-iPSCs have overexpressed OCT4, and a lower than normal WT level of OCT4 actually is beneficial for pluripotency. The authors have to do more analysis and strengthen their claim on the potential mechanistic action of ZNF207 in promoting hiPSC reprogramming.

Response: We agree with the reviewer that the endogenous *OCT4* level must be tightly controlled in hESCs and in the process of reprogramming in order to maintain or establish pluripotency. We think there is a misunderstanding on this point and thus, we have clarified the context and extent of our experiments. We measured endogenous *OCT4* expression in the mixture of cells, including fibroblasts and reprogrammed iPSCs. We did not sort OCT4-positive cells (no OCT4-GFP reporter cells were used in our experiments) and measure the expression of *OCT4* at the single cell level. Thus, the level of endogenous *OCT4* expression reported corresponds to the efficiency of reprogramming. The more cells reprogrammed successfully to iPSCs, the higher level of endogenous *OCT4* expression that is observed in our experimental setting. Thus, our results showed that ZNF207 OE is able to significantly increase reprogramming efficiency, resulting in more fully reprogrammed iPSCs. This indicates that endogenous *OCT4* has been activated to an optimal level in ZNF207 OE cells for reprogramming.

We agree that this result of endogenous *OCT4* expression can only show that ZNF207 promotes reprogramming, but is not sufficient to demonstrate the potential molecular mechanism. To better identify potential mechanistic actions of ZNF207 in promoting iPSCs reprogramming, we reported other experiments in which we mutated ZNF207 protein by deleting its DNA binding domain and introduced it into the reprogramming process and

compared results to those with wildtype ZNF207 protein OE and control (no ZNF207 OE) conditions. This demonstrated that mutated ZNF207 is not able to increase reprogramming efficiency (Fig. 2f). This result suggests that ZNF207 promotes reprogramming by regulation of gene transcription through its DNA binding domain. Our further experiments also demonstrated that ZNF207 binds to the enhancers of *OCT4* (Fig. 2c, 3e) and directly regulates the transcription through these regulatory elements (Supplementary Fig. 4a). Furthermore, the effects of ZNF207 KD were rescued by OE of *OCT4* (Supplementary Fig. 4b, c), indicative of *OCT4* as one of the major downstream targets of ZNF207. Taken together, we propose that ZNF207 promotes reprogramming through regulation of gene expression in a role of DNA binding transcription factor, especially through regulation of key pluripotency genes, including *OCT4*.

To clarify our data, we moved the previous Fig. 2f, which is the measurement of endogenous *OCT4* level to the new Supplementary Fig. 2e and we included new data on mutation of ZNF207 protein in the process of reprogramming as the new Fig. 2f. In the main text, we highlighted our change of text on page 6 as shown below:

“We observed higher expression of endogenous *OCT4* in ZNF207 OE cells and lower expression of endogenous *OCT4* in ZNF207 KD cells, which correlates with the reprogramming efficiency in each condition (Supplementary Fig. 2e). To better understand the molecular mechanism of ZNF207 in promoting the reprogramming efficiency, we mutated ZNF207 protein by deleting its DNA binding domain (ZNF207/MUT) and introduced it to the reprogramming process, in comparison to the wild type ZNF207 protein OE (ZNF207/WT) and no ZNF207 OE (Control) conditions. We found that mutated ZNF207 is not able to promote increased reprogramming efficiency (Fig. 2f). This result suggests that ZNF207 promotes reprogramming by regulation of gene transcription through its DNA binding domain. Taken together, these results indicate that ZNF207 is essential for the maintenance of hESCs and reprogramming of human pluripotent stem cells from somatic cells.”

To further clarify the data, we also moved the description of Supplementary Fig. 4a-c, which shows that *OCT4* is one of major downstream targets of ZNF207, to the next section of the results “Identification of direct ZNF207 target genes in hESCs indicates *OCT4* is one of the direct targets of ZNF207”. In this way, the second section of the results focus on the discussion of the role of ZNF207 in hESC maintenance and iPSCs reprogramming, while the third section of the result focus on the identification of downstream targets of ZNF207. The related text has been changed and highlighted on page 6-7 of the main text.

2. Fig. S2b: HDAC2 and SALL2 siRNA treatment resulted in over 2-fold increase of *OCT4* levels. Since this will cause differentiation of mouse ESCs, what happened to hESCs here should be pointed out.

Response: We thank the reviewer for pointing out these results. The level of *OCT4* expression must be tightly controlled in order to maintain self-renewal and pluripotency. A significant reduction or increase of *OCT4* results in differentiation. We described the KD effects on page 5 of the main text, highlighted in yellow.

“KD of *MBD3*, *CBX3*, *IGF2BP1* and *OCT4* caused reduction of *OCT4* expression, while KD of *HDAC2* and *SALL2* resulted in upregulation of *OCT4* expression (Supplementary Fig. 2b). Both downregulation and upregulation of *OCT4* in hESCs results in differentiation. These results,

consistent with the published data, suggest that *OCT4* level must be tightly controlled in order to maintain self-renewal and pluripotency^{6,23.}”

3. Fig. S3: 1) due to the various cocktails for human naïve ESCs, the authors may want to point out in the main text which cocktail they used for the study shown in Fig. S3; 2) The characterization of KD phenotype is quite superficial and limited with only two marker genes *NANOG* and *SALL4*; 3) Fig. S3c should be supplemented with the *OCT4* expression level to show how much *OCT4* OE is as tight control of *OCT4* level is critical for pluripotency.

Response:

1) We described the strategy we used to derive and maintain naïve pluripotent stem cells on page 6 of the main text.

“We used small molecules to derive and maintain naïve human pluripotent stem cells based on the protocol of Gafni and colleagues^{18.}”

2) In addition to the pluripotency marker genes, we present data on upregulation of lineage marker genes to document differentiation of the cells. The new data is presented in Supplementary Fig. 3c. Taken together, to show the naïve state cells are differentiated in response to *ZNF207* KD, we have presented a set of data, including:

- 1) Disruption of colony forming morphology (Supplementary Fig. 3a);
- 2) Loss of alkaline phosphatase staining (Supplementary Fig. 3b);
- 3) Reduced expression of pluripotency genes and upregulation of lineage marker genes (Supplementary Fig. 3c).

We believe that this set of data is sufficient to draw our conclusion that KD of *ZNF207* causes differentiation of naïve human pluripotent stem cells. We revised the text on page 6 of the main text as follow:

“We observed that depletion of *ZNF207* in naïve-state hPSCs resulted in disruption of colony forming morphology (Supplementary Fig. 3a), loss of alkaline phosphatase staining (Supplementary Fig. 3b), reduced expression of pluripotency genes and upregulation of lineage marker genes (Supplementary Fig. 3c).”

3) *OCT4* levels are now provided in Supplementary Fig. 4c. We titrated the level of *OCT4* to be comparable to its level in hESCs.

4. Fig. S6c: I failed to understand or appreciate the authors’ statement “forced expression of *ZNF207* increased the expression of *OTX2*” based on the bar chart shown. I did not see increased *OTX2* expression at all compared with the Control.

Response: We apologize for the misunderstanding on these data. The KD and OE (should be “rescue by *ZNF207* OE”) experiments were not done on hESCs, but in the process of ectoderm differentiation. In these specific experiments, we differentiated hESCs to ectoderm by subjecting the cells to ectoderm specific media. In “Control”, which is hESCs, cells start to differentiate towards ectoderm and endogenous *OTX2* expression is induced. In “*ZNF207* KD”, we knocked down *ZNF207* utilizing siRNAs and differentiated the cells to ectoderm. In this case, *OTX2* expression is not properly induced, which leads to deficiency of the cells to differentiate to

ectoderm. In “Rescue by ZNF207 OE”, we rescued the ZNF207 KD cells by forced expression of ZNF207 to the endogenous level. In this rescue experiment, we observed an induction of *OTX2* expression again, suggesting that *OTX2* is one of the direct targets of ZNF207 in the process of ectoderm differentiation. We have changed the label of Supplementary Fig. 6c and the text on page 10 of the main text, highlighted in yellow:

“Notably, we observed that *OTX2*, an early ectodermal lineage marker gene, is bound by ZNF207 (Supplementary Fig. 6b). In the process of ectoderm differentiation of hESCs, *OTX2* expression is induced to a proper level to facilitate further differentiation. In ZNF207 KD cells, *OTX2* expression cannot be induced efficiently and is at much lower level compared to the control. However, its expression is restored by a rescue experiment with ZNF207 OE (Supplementary Fig. 6c). These data indicate that *OTX2* is a direct target of ZNF207.”

5. Fig. S6d: the *OTX2* expression level should also be shown to show its overexpression relative to Control.

Response: *OTX2* expression level is now provided in Supplementary Fig. S6d. We expressed *OTX2* to a level comparable to the control. We also note that on page 10 of the main text:

“To address the possibility that reduced expression of *OTX2* might be linked to the deficiency in ectoderm differentiation, we overexpressed *OTX2* in ZNF207 KD hESCs to the comparable level of control cells (Supplementary Fig. 6d) to test whether this could rescue differentiation to ectoderm.”

Minor points:

1. Line 166: KD of ZNF207 KD should be corrected as “KD of ZNF207” or simply “ZNF207 KD”.

Response: We changed it to ZNF207 KD on page 7 of the main text.

2. Line 176: why suddenly hPGCs? Must be the typo for “hPSCs”.

Response: We corrected it as “hPSCs” on page 6 of the main text.

3. Lin 188: After briefly described the KD effects on naïve hESCs above, the authors may want to point out the RNA-seq and CHIP-seq afterwards were also performed in conventional primed hESCs.

Response: We pointed out that RNA-Seq and ChIP-Seq were performed on hESCs on page 6-7 of the main text.

4. Fig. 4 legends was a mess. (c) was presented twice, and c/d legends do not match the data shown. It is such a careless mistake during the revision stage.

Response: We corrected the legends on page 15 of the main text.

5. Fig. 4e: p-value should be explicitly labeled in the figure also.

Response: We provided the p-values explicitly in Fig. 4e.

Reviewer #2 (Remarks to the Author):

Overall, the paper has much improved and specifically the newly added data showing the direct binding of Znf207 to its predicted binding motif does strengthen the point the authors are making and do indeed support the idea that the protein could in fact binding DNA directly and could act as a transcription factor. Therefore, the authors have addressed most of my concerns. However, the authors identified three different isoforms and showed that the Isoform C is the main isoform expressed in human ESC and that this isoform is also the only one that is able to rescue the loss of Znf207. Which isoform did the authors test in their EMSA? Did they test all three isoforms and only the C isoform worked or did they only test one, unnamed isoform? This should also be indicated for Figure 2e and 2f, 5h and 5i (did the authors test the other isoforms in these assays?).

Response: We tested isoform C, which is the main isoform in hESCs and the longest isoform, for the EMSA assay. We also used isoform C for overexpression experiments in Fig. 2e, 2f, 5h and 5i. Since we only pointed out the different isoforms in the last section of the results, it is not appropriate to mention which isoform we used for each of the experiments in the main text before the last section of the results. To solve this, we note on page 11 of the main text, which is the last section of the results, that we used the isoform C for all the ZNF207 protein experiments.

“We note that we used the isoform C for all the ZNF207 protein experiments, including EMSA (Fig. 3e) and overexpression (Fig. 2e, 2f, 5h and 5i) analysis.”

We also specified which isoform we used for EMSA and OE experiments in the section of Material and Methods. The corresponding changes are highlighted on page 18 and 20 of the main text.

Reviewer #3 (Remarks to the Author):

The revised version of the manuscript has substantially improved and addresses the key concerns of the reviewers. The authors carried out a substantial amount of additional experiments to corroborate their conclusions. I recommend to accept this version for publication in Nature Communications.

Response: We thank the reviewer for the recommendation.

Reviewers' comments:

Reviewer #1 (Remarks to the Author):

This revision is largely satisfactory except for one thing. Upon my last request for the particular condition for naïve hESCs, the authors now provided the information that the naïve conversion was based on Gafni and Colleagues. However, this could be troublesome or problematic in their argument following the presentation of these data (Fig. S3) in line 188-189:

"These results suggested that ZNF207 may be a critical regulator in human embryo development; thus, we examined whether ZNF207 is expressed in preimplantation human embryos" for the following reasons:

It is well recognized in the field that the naïve condition from Gafni and Colleagues does not closely approximate human preimplantation blastocyst compared to those of the Jaenisch and Smith naïve hESCs (Huang et al., 2014). Ideally, the authors should use the Jaenisch or Smith naïve conditions instead of the Gafni et al. condition.

Huang, K., Maruyama, T., and Fan, G. (2014). The naïve state of human pluripotent stem cells: a synthesis of stem cell and preimplantation embryo transcriptome analyses. *Cell Stem Cell* 15, 410-415.

We thank the editors and the reviewers again for their final comments. In this revised version of manuscript, we have replaced the original naïve state stem cells data that were based on the protocol of Gafni et al., with new data that were generated on cells that were cultured under Dr. Austin Smith's protocol (Takashima et al., 2014). The new data is presented in Supplementary Fig. 3a-c and we have highlighted the revisions in yellow in the main text. With this final version of the manuscript, we think we have addressed all the reviewers' concerns and we believe that this work is a unique and important piece of work in the field of stem cell biology and contributes to better understanding of the endogenous molecular mechanisms underlying self-renewal and pluripotency. Below is the point-by-point response to the reviewer #1' final comment.

Reviewer #1 (Remarks to the Author):

This revision is largely satisfactory except for one thing. Upon my last request for the particular condition for naïve hESCs, the authors now provided the information that the naïve conversion was based on Gafni and Colleagues. However, this could be troublesome or problematic in their argument following the presentation of these data (Fig. S3) in line 188-189:

“These results suggested that ZNF207 may be a critical regulator in human embryo development; thus, we examined whether ZNF207 is expressed in preimplantation human embryos” for the following reasons:

It is well recognized in the field that the naïve condition from Gafni and Colleagues does not closely approximate human preimplantation blastocyst compared to those of the Jaenisch and Smith naïve hESCs (Huang et al., 2014). Ideally, the authors should use the Jaenisch or Smith naïve conditions instead of the Gafni et al. condition.

Huang, K., Maruyama, T., and Fan, G. (2014). The naïve state of human pluripotent stem cells: a synthesis of stem cell and preimplantation embryo transcriptome analyses. Cell Stem Cell 15, 410-415.

Response: We thank Reviewer #1 for this helpful comment. We have obtained the naïve state stem cells that were derived and cultured under Dr. Austin Smith's protocol (Takashima et al., 2014) from Dr. Sarita P. Panula (Collier et al., 2017). We have performed ZNF207 KD on these naïve state stem cells and we found that depletion of ZNF207 in this naïve-state hPSCs resulted in disruption of colony forming morphology (Supplementary Fig. 3a), loss of alkaline phosphatase staining (Supplementary Fig. 3b), reduced expression of pluripotency genes and upregulation of lineage marker genes (Supplementary Fig. 3c). The new data is shown in Supplementary Fig. 3a-c.

In the main text, we described the strategy that we used to maintain naïve pluripotent stem cells on page 6 of the main text.

“The naïve human pluripotent stem cells were derived and maintained based on the protocol of Takashima et al.,²⁵”

In the “Material and Methods”, we described the source and maintenance of the cells on page 17 of the main text.

“Naïve hPSCs were kindly provided by Dr. Sarita P. Panula⁵⁰ and cultured as previously described²⁵.”

We also included the new references as ref 25 and ref 50 on page 23-24 of the main text.

- 25 Takashima, Y. *et al.* Resetting transcription factor control circuitry toward ground-state pluripotency in human. *Cell* **158**, 1254-1269 (2014).
- 50 Collier, A. J. *et al.* Comprehensive Cell Surface Protein Profiling Identifies Specific Markers of Human Naive and Primed Pluripotent States. *Cell Stem Cell* **20**, 874-890 (2017).